# Amyloid β oligomers suppress excitatory transmitter release via presynaptic depletion of phosphatidylinositol-4,5-bisphosphate

Yang He[1], Mengdi Wei[1], Yan Wu[1], Huaping Qin[1], Weinan Li[1], Xiaolin Ma[1], Jingjing Cheng[1], Jinshuai Ren[1], Ye Shen[1], Zhong Chen[2], Binggui Sun [1], Fu-De Huang[3], Yi Shen [1] & Yu-Dong Zhou [1]

Amyloid β (Aβ) oligomer-induced aberrant neurotransmitter release is proposed to be a crucial early event leading to synapse dysfunction in Alzheimer's disease (AD). In the present study, we report that the release probability (Pr) at the synapse between the Schaffer collateral (SC) and CA1 pyramidal neurons is significantly reduced at an early stage in mouse models of AD with elevated Aβ production. High nanomolar synthetic oligomeric Aβ$_{42}$ also suppresses Pr at the SC-CA1 synapse in wild-type mice. This Aβ-induced suppression of Pr is mainly due to an mGluR5-mediated depletion of phosphatidylinositol-4,5-bisphosphate (PIP$_2$) in axons. Selectively inhibiting Aβ-induced PIP$_2$ hydrolysis in the CA3 region of the hippocampus strongly prevents oligomeric Aβ-induced suppression of Pr at the SC-CA1 synapse and rescues synaptic and spatial learning and memory deficits in APP/PS1 mice. These results first reveal the presynaptic mGluR5-PIP$_2$ pathway whereby oligomeric Aβ induces early synaptic deficits in AD.

[1] Department of Neurobiology, and Department of Ophthalmology of the First Affiliated Hospital, NHC and CAMS Key Laboratory of Medical Neurobiology, Zhejiang University School of Medicine, Hangzhou 310058, China. [2] Department of Pharmacology, NHC and CAMS Key Laboratory of Medical Neurobiology, College of Pharmaceutical Sciences, Zhejiang University, Hangzhou, Zhejiang 310058, China. [3] Center for Stem Cell and NanoMedicine, Laboratory for System Biology, Shanghai Advanced Research Institute, Chinese Academy of Sciences, Shanghai 201210, China. These authors contributed equally: Yang He, Mengdi Wei. Correspondence and requests for materials should be addressed to F.-D.H. (email: huangfd@sari.ac.cn) or to Y.S. (email: yshen2@zju.edu.cn) or to Y.-D.Z. (email: yudongzhou@zju.edu.cn)

The neuropathology of Alzheimer's disease (AD) is characterized by occurrence of senile plaques containing amyloid β (Aβ) aggregates and neurofibrillary tangles formed by hyperphosphorylated tau in the brain[1–3]. An important cellular correlate of cognitive decline in AD is synapse loss[4,5]. Thus, many studies in AD focus on exploring the underlying mechanisms of neurotoxic effects of Aβ and hyperphosphorylated tau on synapse loss and neuronal death[6]. However, synaptic dysfunction may occur before synapse loss in early AD[4]. Therefore, elucidating how pathogenic Aβ and tau species alter synaptic transmission is crucial to the diagnosis and treatment of AD. In recent years, investigating how Aβ modulates synapse function in early AD has attracted great attention[7,8]. The neurotoxic soluble Aβ oligomers, including the most toxic oligomeric Aβ42, have been shown to alter synaptic plasticity and synaptic transmission in various AD animal models via a variety of synaptic targets of Aβ such as ionic neurotransmitter receptors, G protein-coupled receptors (GPCRs), receptor tyrosine kinases, and cellular prion proteins (PrP$^C$)[9,10]. Although many of these Aβ targets exist in both presynaptic and postsynaptic loci[11], and Aβ oligomers accumulate at both sides of the excitatory synapse[12,13], majority of the studies have only examined the toxic gain of function for Aβ as a result of its interaction with the postsynaptic targets[14,15]. Examinations of Aβ-induced abnormalities in synaptic transmission have nevertheless revealed presynaptic defects are often more prominent than postsynaptic abnormalities[16,17]. Thus, it is pivotal to unravel the presynaptic targets of Aβ in AD.

Physiological concentration of Aβ (picomolar)[18] has been shown to positively regulate synaptic transmission via upregulating the presynaptic neurotransmitter release probability (Pr)[19]. Low to moderate levels of Aβ may augment Pr via increasing presynaptic Ca$^{2+}$ by promoting presynaptic amyloid precursor protein (APP) homodimerization[20], activating exocytotic Ca$^{2+}$ channels[21], and regulating presynaptic α7 nicotinic acetylcholine receptors[22,23]. However, how pathological level of oligomeric Aβ (nanomolar)[18] leads to presynaptic defects remains largely obscure. Controversial results exist in the literature showing pathogenic Aβ may exert negative[16,17,24–26], positive[27,28], or no[29,30] effects on neurotransmitter release. Different Aβ species and their targets, various assemblies of Aβ monomers, and duration of Aβ action may account for these disparate observations[9]. Furthermore, there is little evidence of what the presynaptic targets of pathogenic Aβ are in early AD. There are contradicting results with respect to how nanomolar Aβ oligomers regulate voltage-gated Ca$^{2+}$ channels[25,27,31] and SNARE complex proteins[26,28] to disrupt presynaptic neurotransmitter release.

In the current study, we aim to clarify the presynaptic deficit at an excitatory hippocampal synapse in AD models and determine the presynaptic target of pathological level of oligomeric Aβ42. We identified a nanomolar oligomeric Aβ42-induced, presynaptic metabotropic glutamate receptor 5 (mGluR5)-mediated hydrolysis of membrane phosphatidylinositol-4,5-bisphosphate (PIP$_2$) underlies the diminished Pr in early AD. Postsynaptic mGluR5 has been shown to function as an Aβ receptor or co-receptor with PrP$^C$[32–34] and blocking mGluR5 reduces cognitive impairment in AD mouse models[35–37]. Oligomeric Aβ[38] and apolipoprotein E4[39] are also known to interfere with PIP$_2$ metabolism and reduction of PIP$_2$ phosphatase synaptojanin 1 ameliorates synaptic and behavioral deficits in AD[40]. Our results for the first time establish that increasing the presynaptic PIP$_2$ level is an effective way to improve cognition in AD.

## Results

### Reduced transmission in early AD involves a reduction in Pr.
To determine the synaptic deficits in early AD, we first examined spine morphology and density of apical dendrites in CA1 pyramidal

neurons in 6–7-month-old APP (Swe); PS1(ΔE9) (APP/PS1) mice (Fig. 1a–c). We compared the total spine density (Fig. 1c) and the density of filopodium-like, thin, stubby, and mushroom-shaped spines (Fig. 1b) in wild-type (WT) and APP/PS1 mice. We found that in APP/PS1 mice, there were no changes in the total spine density and the density of mature forms of spines in comparison to WT mice, although the filopodium-like spine density was significantly increased. The result is consistent with a previous study showing spine loss did not occur in CA1 stratum radiatum in APP/PS1 mice, whereas the number of small-headed spines was significantly increased[41]. We next recorded miniature excitatory postsynaptic currents (mEPSCs) in CA1 pyramidal neurons in 6–7-month-old WT and APP/PS1 mice (Fig. 1d, e). The frequency but not the amplitude of mEPSCs was significantly lower in APP/PS1 mice than in WT controls. We then recorded evoked EPSCs at the Shaffer collateral (SC) to CA1 pyramidal neuron (SC-CA1) synapse and found that the amplitude of SC-CA1 EPSCs was strongly decreased in 6–7-month-old APP/PS1 mice (Fig. 1f, g). These changes in mEPSC frequency and SC-CA1 EPSC amplitude were not observed in 4-month-old APP/PS1 mice (Supplementary Fig. 1), suggesting that the synaptic transmission deficits occur at a later stage when Aβ production is greatly elevated (Supplementary Fig. 2). Indeed, in 4-month-old APP (Swe/Flo/Lon); PS1 (M146L/L286V) mice (5xFAD) with an accelerated Aβ production (Supplementary Fig. 2), the frequency of mEPSCs in CA1 neurons and the amplitude of SC-CA1 EPSCs were greatly reduced in comparison to APP/PS1 and WT mice of the same age (Supplementary Fig. 1a–d). By contrast, 6–7-month-old PS1$^{M146V}$ knock-in mice (M146V) with a low Aβ production exhibited slightly decreased mEPSC frequency and amplitude compared to their WT littermates (Supplementary Fig. 1e, f). Interestingly, SC-CA1 EPSCs were smaller in M146V than in control mice at 6–7-month of age, implying that an Aβ-independent mechanism may underlie the decreased synaptic transmission. These results suggest that decreased synaptic transmission in the hippocampus may be a hallmark in AD mouse models that show a high level of Aβ accumulation.

To further test if the reduced hippocampal synaptic transmission in AD mouse models is dependent on Aβ levels, we first bath applied synthetic oligomeric Aβ42 at various concentrations in brain slices from WT mice. We observed that bath application of 400 nM oligomeric Aβ42 for 20 min induced a significant decrease in the amplitude of SC-CA1 EPSCs, whereas 20 nM and 100 nM oligomeric Aβ42 respectively exerted positive and no effects on the evoked responses (Supplementary Fig. 3). Four hundred nanometers of Aβ42 fibrils, on the other hand, did not alter the amplitude of SC-CA1 EPSCs (Supplementary Fig. 3). We thus chose 400 nM Aβ42 oligomers for further in vitro studies. Bath application of 400 nM oligomeric Aβ42 for 20 min reduced both mEPSC amplitude and frequency in CA1 pyramidal neurons (Fig. 1h, i) and suppressed SC-CA1 EPSC amplitude (Fig. 1j, k) in WT mice. Interestingly, 400 nM oligomeric Aβ42 induced a greater suppression of mEPSCs in frequency than in amplitude (Fig. 1e), suggesting that the site of action of oligomeric Aβ42 was mainly presynaptic. These data indicate that 400 nM oligomeric Aβ42 exerts similar effect on hippocampal synaptic transmission as in AD mouse models with a high level of Aβ accumulation.

Although there was no spine loss in 6–7-month-old APP/PS1 mice, Aβ-induced reduction of the number of active boutons may account for the decreased mEPSC frequency in these animals, as nanomolar Aβ oligomers can reduce the recycling pool and increase the resting pool of synaptic vesicles[42]. To test this hypothesis, we loaded active synaptic vesicles with FM1–43 in cultured hippocampal neurons. Compared to vehicle treatment, treatment with 400 nM oligomeric Aβ42 did not change the number of active boutons in hippocampal neurons (Fig. 1l, m). These results indicate that Aβ-induced inhibition of mEPSC

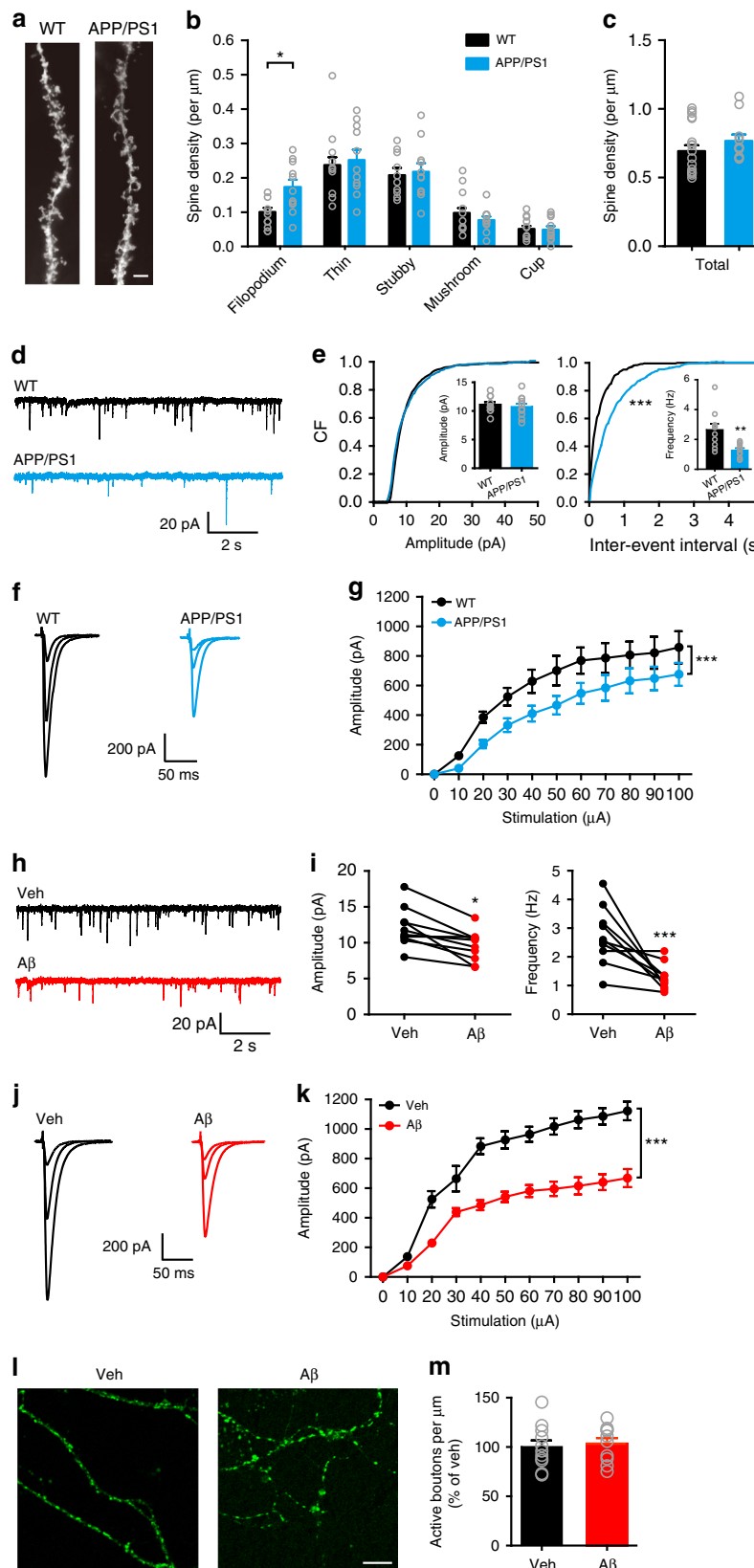

frequency is not due to a decrease in the number of active boutons and suggest that early synaptic deficits in AD mouse models may involve a presynaptic mechanism other than reducing the active pool of synaptic vesicles.

A decrease in mEPSC frequency may also be due to a reduction in presynaptic release Pr. We thus first investigated whether Aβ

elevation enhances paired-pulse facilitation (PPF) at the SC-CA1 synapse, as the degree of PPF is inversely related to Pr. We found that PPF was significantly increased in 6–7-month-old APP/PS1 (Fig. 2a, b) and 4-month-old 5xFAD (Supplementary Fig. 4a, b) mice compared to WT controls. In 4-month-old APP/PS1 (Supplementary Fig. 4a, b) and 6–7-month-old M146V

**Fig. 1** Excitatory synaptic deficits in 6–7-month-old APP/PS1 mice involve a presynaptic mechanism independent of altered bouton density. **a–c** Representative Golgi staining of apical dendrites (**a**) and quantification of density of different types of spines (**b**) and total spine density (**c**) in CA1 pyramidal neurons in WT and APP/PS1 mice. Bar, 5 μm. *t* test; *, $P < 0.05$; $N = 10$–19 per group. **d, e** Representative traces (**d**) of mEPSCs in CA1 pyramidal neurons and cumulative plots and mean values (insets) (**e**) of mEPSC amplitude (left) and frequency (right) in WT and APP/PS1 mice. Kolmogorov-Smirnov test (cumulative plots), ***$P < 0.001$; *t* test (insets), **$P < 0.01$; $N = 10$–11 per group. **f, g** Representative traces of SC-CA1 EPSCs evoked by stimulus intensities of 10, 30, and 100 μA (**f**) and quantification of EPSC amplitude to stimulus intensity (**g**) in WT and APP/PS1 mice. Two-way ANOVA with post hoc Bonferroni test; animal, $F_{(1,132)} = 30.22$, $P < 0.001$; stimulation, $F_{(10,132)} = 27.23$, $P < 0.001$; ***$P < 0.001$; $N = 5$–9 per group. **h, i** Representative traces (**h**) of mEPSCs in CA1 pyramidal neurons and quantification (**i**) of mEPSC amplitude (left) and frequency (right) in WT hippocampal slices before (vehicle, Veh) and after Aβ treatment. *t* test; *$P < 0.05$; ***$P < 0.001$; $N = 10$ per group. **j, k** Representative traces of SC-CA1 EPSCs evoked by stimulus intensities of 10, 20, 100 μA (**j**) and quantification of EPSC amplitude to stimulus intensity (**g**) in WT hippocampal slices before (Veh) and after Aβ treatment. Two-way ANOVA with post hoc Bonferroni test; animal, $F_{(1,132)} = 244.0$, $P < 0.001$; stimulation, $F_{(10,132)} = 83.89$, $P < 0.001$; ***$P < 0.001$; $N = 7$ per group. **l, m** Representative images of FM1–43-labeled active boutons (**l**) and quantification of relative FM1–43-labeled bouton density (**m**) in cultured hippocampal neurons treated with Veh or Aβ. Bar, 50 μm. *t* test; $P > 0.05$; $N = 10$–11 per group. Data are mean ± SEM. CF cumulative frequency. Source data are provided as a Source Data file

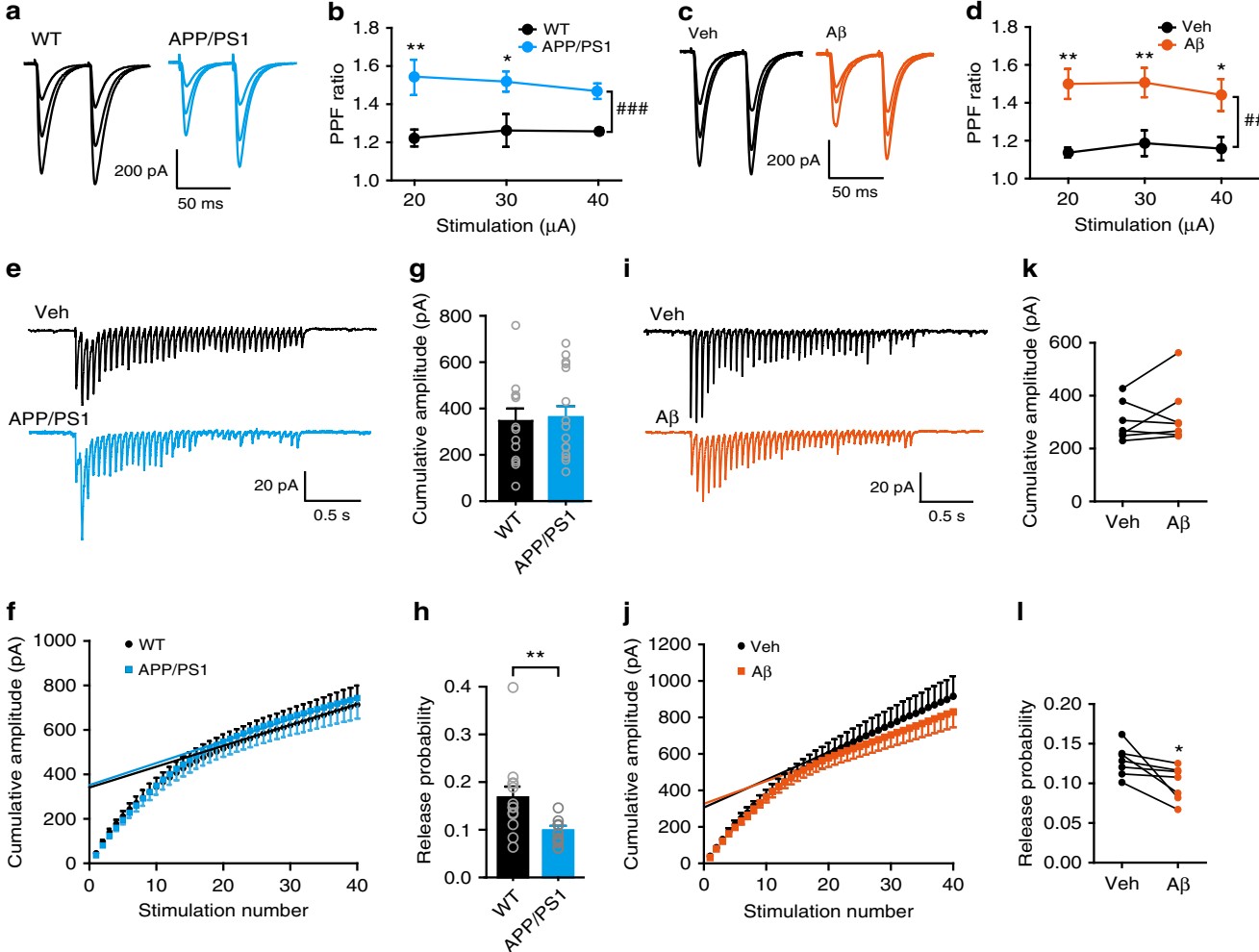

**Fig. 2** High nanomolar Aβ oligomers suppress presynaptic release probability at the SC-CA1 synapse. **a–d** Representative traces (**a**, **c**) and quantification (**b**, **d**) of PPF of SC-CA1 EPSCs evoked by stimulus intensities of 20, 30, and 40 μA in 6–7-month-old WT and APP/PS1 mice (**a**, **b**) and in oligomeric Aβ (400 nM)-treated hippocampal slices relative to vehicle (Veh)-treated ones (**c**, **d**). Two-way ANOVA with post hoc Bonferroni test; in **b**: animal, $F_{(1,24)} = 32.67$, $P < 0.001$; stimulation, $F_{(2,24)} = 0.2378$, $P = 0.89$; in **d**: treatment, $F_{(1,24)} = 34.63$, $P < 0.001$; stimulation, $F_{(2,24)} = 0.029$, $P = 0.79$; ###$P < 0.001$; *$P < 0.05$; **$P < 0.01$; $N = 5$ per group. **e** Representative traces showing the response of CA1 pyramidal neurons to 20 Hz stimulation of the SC in 6–7-month-old WT and APP/PS1 mice. **f–h** Cumulative amplitude analysis showing the magnitude of the cumulative amplitude (**f**), RRP size (**f**, **g**), defined as the y-intercept of the linear portion of the curve, and the release probability (**h**), calculated as mean EPSC amplitude (the mean value of the 1st EPSCs) divided by RRP size, in WT and APP/PS1 mice. *t* test; **$P < 0.05$; $N = 13$–15 per group. **i** Representative traces showing the response of the SC-CA1 synapse to 20 Hz stimulation before (Veh) and after bath application of 400 nM Aβ oligomers in WT hippocampal slices. **j–l** Cumulative amplitude analysis (**j**) showing the effect of 400 nM Aβ oligomers on RRP size (**j**, **k**) and the release probability (**l**) at the SC-CA1 synapse. *t* test; *$P < 0.05$; $N = 7$ per group. Data are mean ± SEM. Source data are provided as a Source Data file

(Supplementary Fig. 4c, d) mice, however, PPF was unchanged in comparison to WT mice, which is also reported in 3-month-old APP/PS1 mice[43]. Similarly, $A\beta_{42}$ oligomers induced a significant increase in PPF in WT animals (Fig. 2c, d). These results suggest that pathogenic $A\beta$-induced synaptic deficits are due to a reduction in Pr. To directly determine if Pr was suppressed in APP/PS1 mice and in $A\beta$-treated brain slices from WT mice, we used a repeated stimulation protocol to estimate the readily releasable pool (RRP) size and Pr (Fig. 2e–l). Repeated stimulation (20 Hz) revealed a significant reduction in Pr in APP/PS1 mice (Fig. 2h) and in $A\beta$-treated brain slices from WT mice (Fig. 2l), although the RRP size did not change (Fig. 2g, k). Taken together, these results indicate that $A\beta$ accumulation reduces Pr in the hippocampus.

To establish that the presynaptic deficit is due to an elevated $A\beta$ level in 6–7-month-old APP/PS1 mice, we investigated whether reducing the $A\beta$ level could restore hippocampal synaptic transmission. We first treated 6–7-month-old APP/PS1 and WT mice with LY-411575, a γ-secretase inhibitor that is known to decrease interstitial fluid levels of $A\beta$[18,44], and confirmed the effectiveness of LY-411575 by the elevated levels of C-terminal fragments (CTFs) after treatment (Supplementary Fig. 5a). We did not measure $A\beta$ levels to indicate LY-411575 activity because existing plaques in these mice are not reduced by LY-411575[44,45]. Treatment with LY-411575 significantly increased CA1 mEPSC frequency in CA1 pyramidal neurons (Supplementary Fig. 5b, c) and SC-CA1 EPSC amplitude (Supplementary Fig. 5d, e) and reduced PPF at the SC-CA1 synapse (Supplementary Fig. 5f, g) in APP/PS1 mice, implicating that inhibiting $A\beta$ generation enhanced Pr in these mice. We next examined whether directly blocking $A\beta$ with a specific human $A\beta$ antibody could restore the decreased Pr in APP/PS1 mice. Long-term incubating hippocampal slices from APP/PS1 mice with 6E10 (2 μg ml$^{-1}$, >5 h) strongly increased mEPSC frequency (Fig. 3a, b) and evoked EPSC amplitude (Fig. 3c, d) and suppressed PPF (Fig. 3e, f), indicating that blocking $A\beta$ with 6E10 indeed restored the presynaptic defect in these mice. Although this long-term maintenance of hippocampal slices in vitro caused a rundown in the baseline transmission in both WT and APP/PS1 mice, the effect of blocking $A\beta$ with 6E10 on restoring synaptic transmission in APP/PS1 mice to the level in WT mice was remarkable. Taken together, our results showed that elevated $A\beta$ levels were essential for reducing Pr in APP/PS1 mice.

**PIP$_2$ depletion by $A\beta$-induced activation of mGluR5 reduces Pr.** One of the key mechanisms controlling Pr involves a vesicle membrane-anchoring event preceding the formation of the SNARE complex. As a crucial phosphoinositide interacting with membrane-binding proteins, PIP$_2$ has been shown to play an important role in vesicle release via synaptotagmin 1-PIP$_2$ binding, and its level in the membrane, thus, is tightly associated with Pr[46]. We found that the PIP$_2$ level was significantly decreased in 6–7-month-old APP/PS1 mice (Supplementary Fig. 6). This is consistent with the decreased Pr in AD mouse models. To further test if $A\beta$ can deplete PIP$_2$ in axons, we immunostained and biochemically measured PIP$_2$ in cultured hippocampal neurons in control, vehicle (DMSO)-treated, or $A\beta_{42}$ oligomer-treated medium (Fig. 4a–c). In comparison to control and DMSO treatments, $A\beta$ treatment rapidly and significantly suppressed total PIP$_2$ (Fig. 4c) and axonal (MAP2$^-$, NF$^+$) and dendritic (MAP2$^+$, NF$^+$) PIP$_2$ (Fig. 4a, b). Although the anti-PIP$_2$ antibody we used cross-reacts with phosphatidylinositol-3,4,5-trisphosphate (PIP$_3$) (Supplementary Fig. 7a), we believe $A\beta$-induced suppression of PIP$_2$ is valid since PIP$_3$ is significantly less abundant than PIP$_2$[47] and the ELISA kit

we used is highly specific to PIP$_2$ (Supplementary Fig. 7b). To test if adding PIP$_2$ back to $A\beta$-treated neurons was sufficient to restore $A\beta$-induced presynaptic defect, we recorded miniature excitatory autaptic currents (mEACs) from individually inhabited neurons grown on collagen-poly-D-lysine (PDL) islands (Fig. 4d). We first established that $A\beta$-induced suppression of mEAC frequency was similar to that of mEPSC frequency (Fig. 4e, f). We then filled the patch pipettes with an internal solution containing 200 μM diC8-PIP$_2$, assuming PIP$_2$ would be incorporated into the axon influencing neurotransmitter release. Indeed, intracellularly applied PIP$_2$ rescued $A\beta$–induced inhibition of mEAC frequency, whereas intracellularly applied PIP$_2$ did not affect baseline mEAC frequency (Fig. 4e, f). These results indicate that $A\beta$-associated PIP$_2$ depletion accounts for $A\beta$-induced presynaptic effect.

$A\beta$-induced rapid PIP$_2$ depletion confirms a predominant PIP$_2$ hydrolysis process evoked by $A\beta$ oligomers[38]. Increased PLC activity, thus, may result in $A\beta$-induced PIP$_2$ depletion. To test this hypothesis, we applied PLC inhibitor U73122 in the culture medium to block PLC activity in primary hippocampal neurons before oligomeric $A\beta_{42}$ treatment. In the presence of U73122, $A\beta_{42}$ oligomers no longer exerted the inhibitory effect on PIP$_2$ levels in axons and dendrites (Fig. 5a, b). U73122 alone did not change PIP$_2$ levels in neuronal processes (Fig. 5a, b). These results indicate that $A\beta$-induced PIP$_2$ depletion is mainly due to an $A\beta$-triggered, PLC-mediated PIP$_2$ hydrolytic event. We next investigated whether blocking PLC activity prevented the presynaptic deficit induced by oligomeric $A\beta_{42}$. In the presence of U73122, oligomeric $A\beta_{42}$-induced suppression of SC-CA1 EPSCs was partially restored (Fig. 5c–e). Notably, U73122 treatment reduced oligomeric $A\beta_{42}$-induced inhibition of mEPSC frequency but not amplitude (Fig. 5f, g), suggesting that blocking PLC prevented $A\beta$-induced inhibition of Pr. Blocking $A\beta$-induced inhibition of Pr via inhibiting PLC was further proved by examining PPF at the SC-CA1 synapse, as a prior application of U73122 prevented $A\beta$-induced increase in the paired-pulse ratio (Fig. 5h, i). In addition, U73122 alone did not change the frequency and amplitude of mEPSCs in CA1 pyramidal neurons and the PPF at the SC-CA1 synapse in WT animals (Supplementary Fig. 8a-d). Furthermore, we explored which PLC isoforms were involved in $A\beta$-induced PIP$_2$ hydrolysis in neurites using the RNA-interference approach (Supplementary Fig. 9a, b). Knocking down either PLCβ1 or β4 significantly ameliorated $A\beta$-induced suppression of PIP$_2$ levels in both dendrites and axons (Supplementary Fig. 9c, d), indicating that both PLCβ1 and β4 contribute to $A\beta$-induced PIP$_2$ hydrolysis in hippocampal neurons. Taken together, these results prove that $A\beta$-induced elevation of PLC activity underlies PIP$_2$ depletion and Pr reduction.

One of the key routes leading to PLC activation is via ligand binding to Gαq-coupled GPCRs. $A\beta$ has been shown to activate many GPCRs, including mGluR5[32–34], a subtype of group I metabotropic glutamate receptors (mGluRs). We found that mGluR5 was expressed in neuronal processes, although axonal mGluR5 is less abundant than dendritic mGluR5 (Fig. 6a, b). Thus, we studied whether mGluR5 mediated $A\beta$-induced PIP$_2$ depletion and the subsequent presynaptic defect. We first examined whether a group I mGluR agonist (S)−3,5-Dihydroxyphenylglycine (DHPG) (Fig. 6) exerted similar effects as $A\beta$ oligomers (Fig. 4). Application of DHPG in the culture medium significantly decreased PIP$_2$ levels in axons and dendrites in primary hippocampal neurons (Fig. 6c, d). Unlike $A\beta$, DHPG still depleted PIP$_2$ in neurites in the presence of an antibody (6D11) against PrP$^C$, indicating that oligomeric $A\beta$-induced PIP$_2$ hydrolysis requires activation of mGluR5 with the aid of PrP$^C$ (Supplementary Fig. 10). Functionally, DHPG inhibited mEPSC

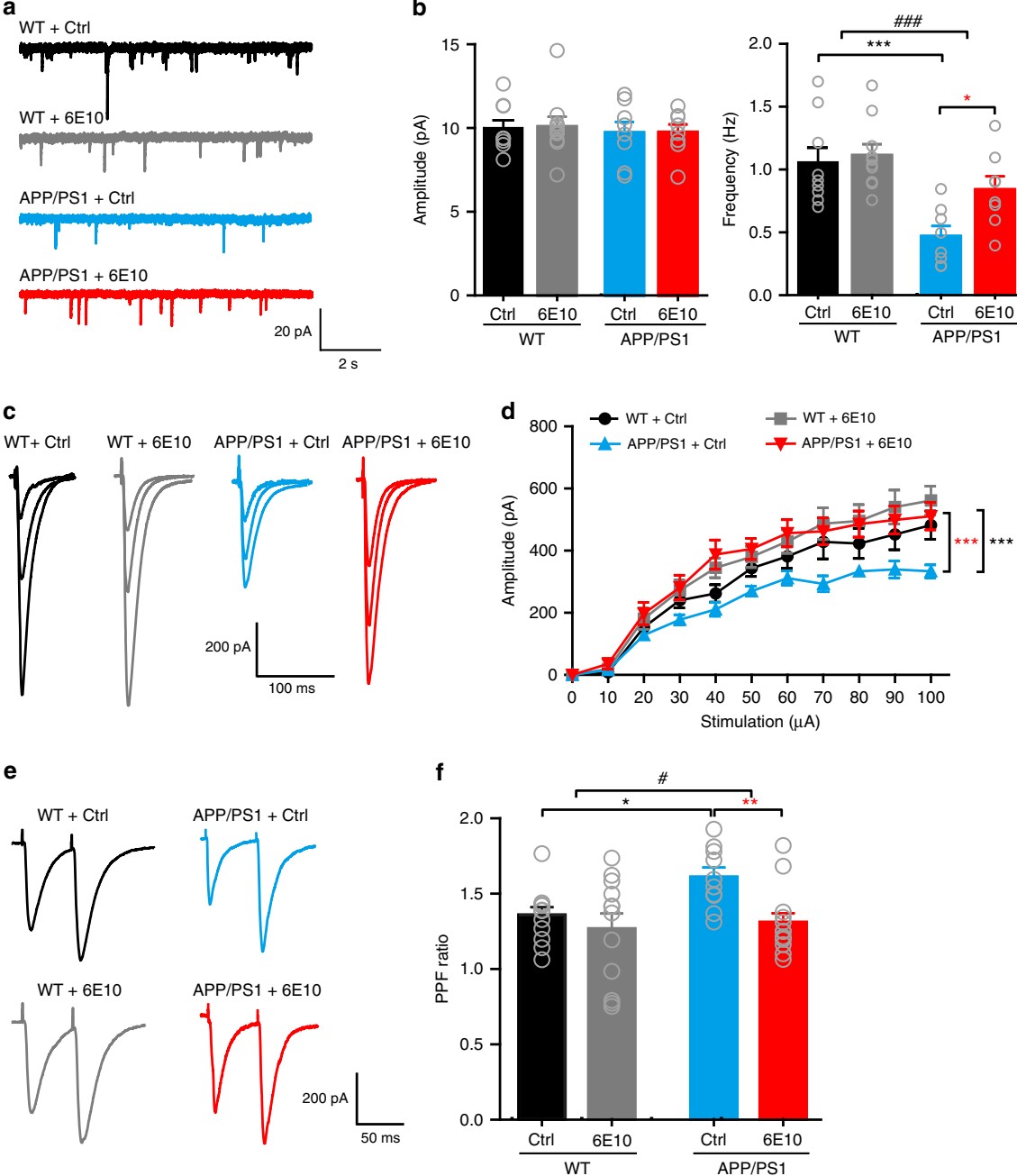

**Fig. 3** Blocking Aβ with anti-β-amyloid antibody 6E10 rescues the presynaptic deficit in hippocampal slices from 6–7-month-old APP/PS1 mice. **a**, **b** Representative traces (**a**) of mEPSCs in CA1 pyramidal neurons and quantification (**b**) of mEPSC amplitude (left) and frequency (right) in hippocampal slices from WT and APP/PS1 mice incubated in control (Ctrl) ACSF or ACSF containing 6E10. Two-way ANOVA with post hoc Bonferroni test; left panel: animal, $F_{(1,31)} = 0.286$, $P = 0.597$; treatment, $F_{(1,31)} = 0.0119$, $P = 0.914$; right panel: animal, $F_{(1,31)} = 18.66$, $P < 0.001$; treatment, $F_{(1,31)} = 4.64$, $P = 0.039$; *$P$ < 0.05; ***$P$ < 0.001; ###$P$ < 0.001 (APP/PS1 vs. WT); $N = 8$–10 per group. **c**, **d** Representative traces of SC-CA1 EPSCs evoked by stimulus intensities of 20, 40, and 100 μA (**c**) and quantification of EPSC amplitude to stimulus intensity (**d**) in hippocampal slices from WT and APP/PS1 mice incubated in Ctrl ACSF or ACSF containing 6E10. Two-way ANOVA with post hoc Bonferroni test; animal, $F_{(3,341)} = 19.694$, $P < 0.001$; treatment, $F_{(10,341)} = 32.140$, $P < 0.001$; ***$P < 0.001$ (compared with APP/PS1 + Ctrl group); $N = 7$–10 per group. **e**, **f** Representative traces (**e**) and quantification (**f**) of PPF of SC-CA1 EPSCs in hippocampal slices from WT and APP/PS1 mice incubated in Ctrl ACSF or ACSF containing 6E10. Two-way ANOVA with post hoc Bonferroni test; animal, $F_{(1,43)} = 4.069$, $P = 0.049$; treatment, $F_{(1,43)} = 7.219$, $P = 0.01$; *$P$ < 0.05; **$P$ < 0.01; #$P$ < 0.05 (APP/PS1 vs. WT); $N = 10$–13 per group. Source data are provided as a Source Data file

frequency in CA1 pyramidal neurons (Fig. 6e, f) and increased PPF at the SC-CA1 synapse (Fig. 6g, h), indicating that activation of group I mGluRs induced mainly a presynaptic effect. Unlike Aβ, DHPG did not alter mEPSC amplitude (Fig. 6e, f), suggesting that activation of group I mGluRs was not involved in Aβ-induced suppression of mEPSC amplitude.

We next investigated whether inhibiting mGluR5 hampered the effectiveness of oligomeric $A\beta_{42}$ on $PIP_2$ depletion and Pr reduction. In the presence of a selective mGluR5 antagonist 3-((2-Methyl-4-thiazolyl)ethynyl)pyridine (MTEP), oligomeric $A\beta_{42}$ no longer suppressed $PIP_2$ levels in neurites in primary hippocampal neurons (Fig. 6i, j). MTEP treatment partially occluded Aβ-

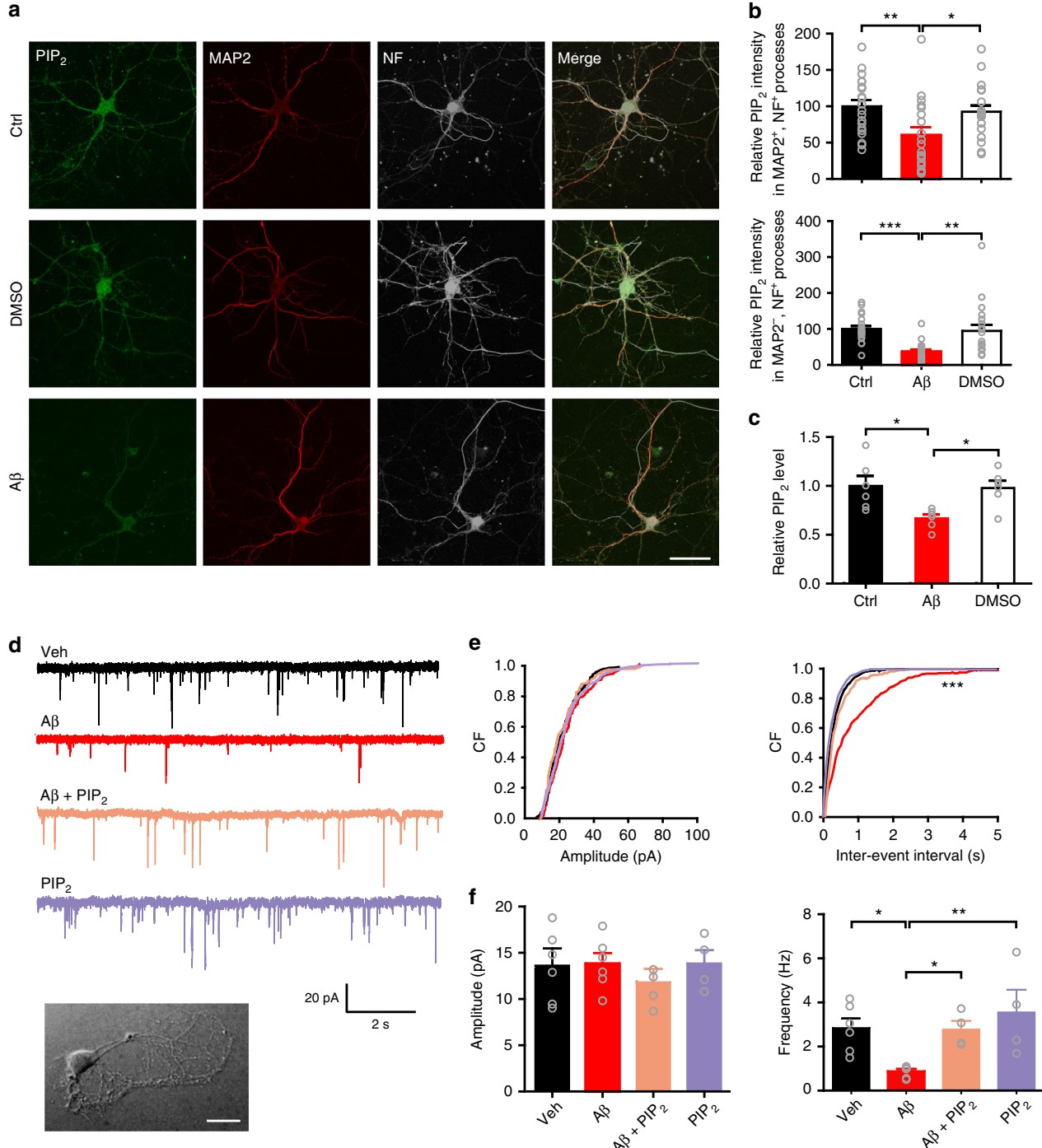

**Fig. 4** Reduced axonal PIP$_2$ accounts for oligomeric Aβ-induced suppression of presynaptic release probability at the SC-CA1 synapse. **a** Confocal images of primary hippocampal neurons showing colocalization of PIP$_2$, MAP2, and neurofilament (NF) along neuronal processes in control, DMSO-treated, and oligomeric Aβ-treated hippocampal neurons. Bar, 50 μm. **b** Histograms showing oligomeric Aβ suppresses PIP$_2$ levels significantly in both dendrites (MAP2$^+$, NF$^+$, upper panel) and axons (MAP2$^-$, NF$^+$, bottom panel). One-way ANOVA with post hoc Dunnett's test; $F_{(2,55)} = 4.95$ (upper); $F_{(2,55)} = 9.39$ (bottom); *$P < 0.05$; **$P < 0.01$; ***$P < 0.001$; $N = 19$–20 per group. **c** Quantification of PIP$_2$ levels measured with ELISA showing oligomeric Aβ suppresses PIP$_2$ in cultured hippocampal neurons. One-way ANOVA with post hoc Dunnett's test; $F_{(2,15)} = 5.87$; *$P < 0.05$; $N = 6$ per group. **d** Representative traces of mEACs recorded from isolated hippocampal neurons (an example shown in inset at bottom) in vehicle-treated medium (Veh), oligomeric Aβ-supplemented medium (Aβ), oligomeric Aβ-supplemented medium with intracellular application of PIP$_2$ (Aβ + PIP$_2$), or vehicle-treated medium with intracellular application of PIP$_2$ (PIP$_2$). Bar, 50 μm. **e**, **f** Cumulative plots (**e**) and mean values (**f**) of mEAC amplitude (left) and frequency (right) in isolated hippocampal neurons in various conditions. Kolmogorov-Smirnov test in **e**; one-way ANOVA with post hoc Dunnett's test in **f**, $F_{(3,16)} = 0.54$ (amplitude); $F_{(3,16)} = 5.47$ (frequency); *$P < 0.05$; **$P < 0.01$; ***$P < 0.001$; $N = 4$–6 per group. Data are mean ± SEM. Source data are provided as a Source Data file

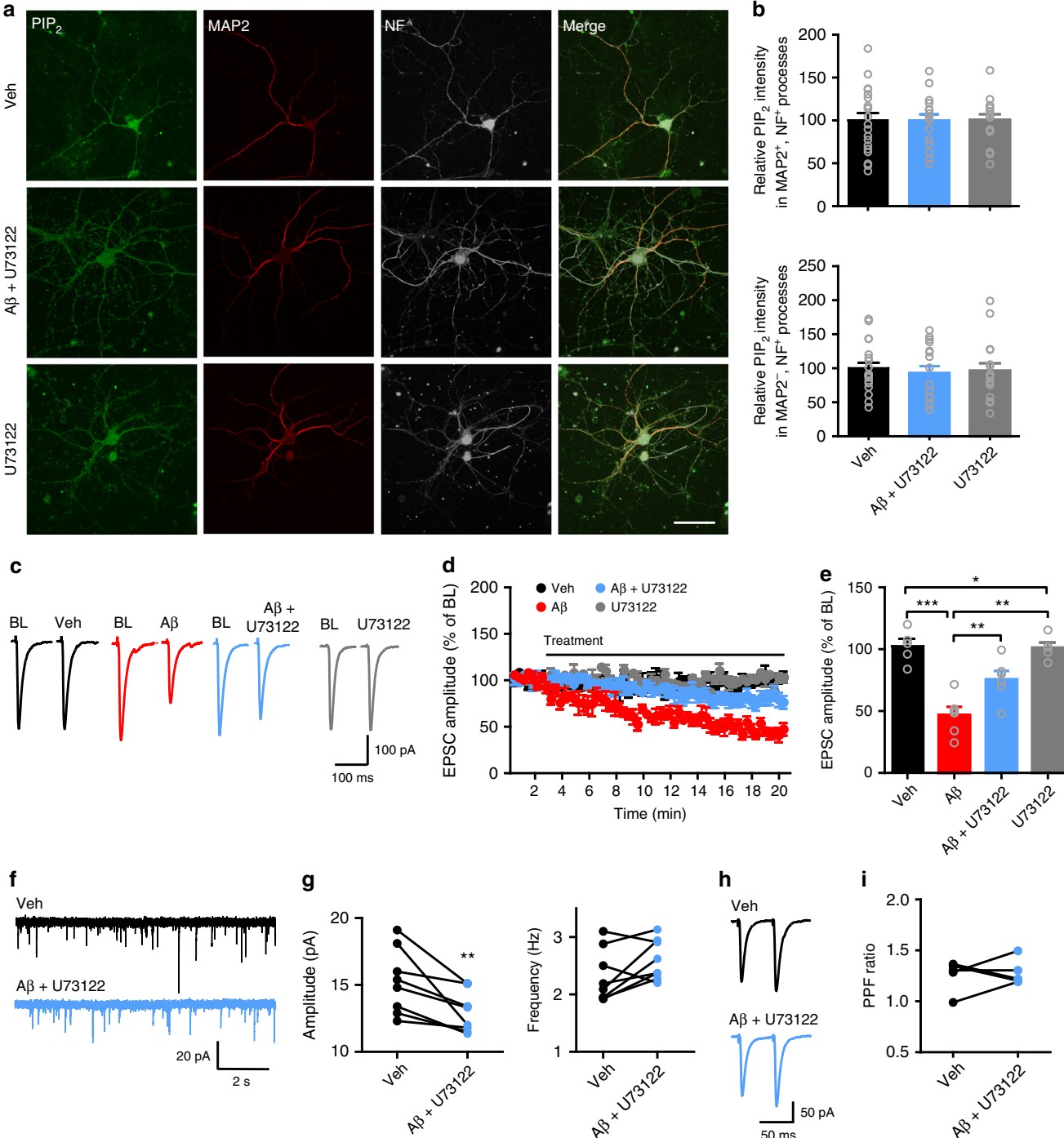

**Fig. 5** Oligomeric Aβ-induced PIP$_2$ reduction and presynaptic deficit are prevented by inhibiting PLC. **a** Confocal images of primary hippocampal neurons showing the effect of oligomeric Aβ on PIP$_2$ levels in neuronal processes in the presence of PLC blocker U73122. Bar, 50 μm. **b** Quantification of relative PIP$_2$ intensity in dendrites (upper panel) and axons (bottom panel) showing U73122 prevents Aβ-induced suppression of PIP$_2$ in neuronal processes. One-way ANOVA with post hoc Dunnett's test; $F_{(2,49)} = 0.007$ (upper); $F_{(2,49)} = 0.12$ (bottom); $P > 0.05$; $N = 16$–20 per group. **c, d** Representative traces (**c**) and the time course of the normalized amplitude (**d**) of SC-CA1 EPSCs in WT hippocampal slices before (baseline, BL) and after Veh, Aβ, Aβ + U73122, or U73122 treatment. **e** Bar graph representing the relative magnitude of EPSCs recorded in the last 1 min of drug treatment shown in **d**. One-way ANOVA with post hoc Dunnett's test; $F_{(3,21)} = 18.04$; *$P < 0.05$; **$P < 0.01$; ***$P < 0.001$; $N = 5$–6 per groups. **f, g** Representative traces (**f**) and quantification (**g**) of mean values of the amplitude (left) and frequency (right) of mEPSCs in CA1 pyramidal neurons before (Veh) and after Aβ + U73122 treatment. $t$ test; **$P < 0.01$; $N = 8$ per group. **h, i** Representative traces (**h**) and quantification (**i**) of PPF of SC-CA1 EPSCs before (Veh) and after Aβ + U73122 treatment. $t$ test; $P > 0.05$; $N = 5$ per group. Data are mean ± SEM. Source data are provided as a Source Data file

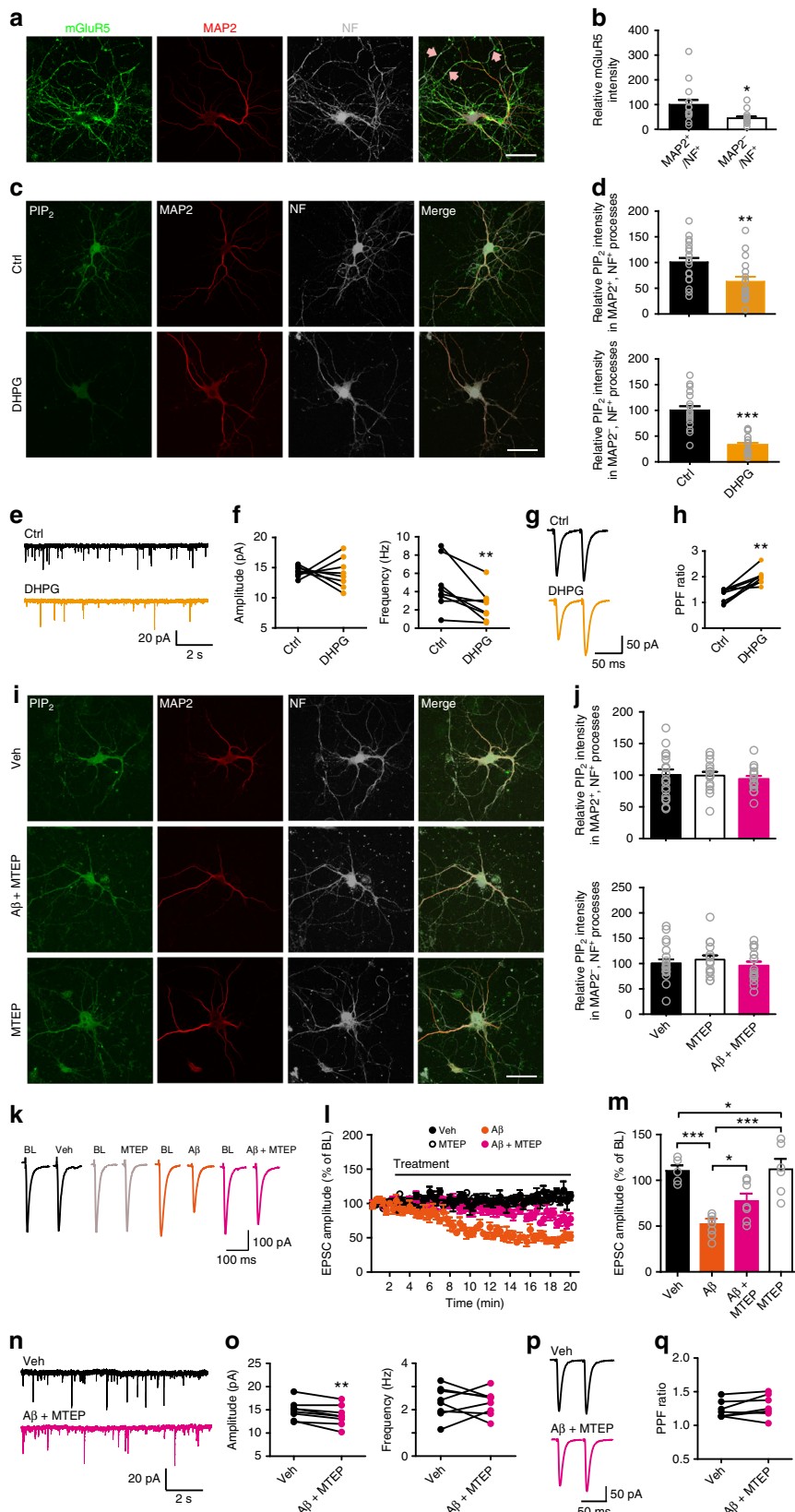

induced inhibition of SC-CA1 EPSCs (Fig. 6k–m), but did not prevent Aβ-induced inhibition of mEPSC amplitude in CA1 pyramidal neurons (Fig. 6n, o), which was consistent with the result showing DHPG did not change mEPSC amplitude (Fig. 6e, f). Importantly, MTEP treatment prevented oligomeric Aβ$_{42}$-

induced changes in mEPSC frequency in CA1 pyramidal neurons (Fig. 6n, o) and PPF at the SC-CA1 synapses (Fig. 6p, q), indicating that blocking mGluR5 ameliorated Aβ-induced Pr suppression. MTEP alone did not alter PIP$_2$ levels in neurites (Fig. 6i, j), SC-CA1 EPSCs (Fig. 6k–m), mEPSCs in CA1

**Fig. 6** Oligomeric Aβ-induced PIP$_2$ reduction and presynaptic deficit involve mGluR5 activation. **a** Confocal images of primary hippocampal neurons showing colocalization of mGluR5, MAP2, and NF along neurites. Arrows: mGluR5 + axons. Bar, 50 μm. **b** Quantification of relative mGluR5 intensity in neurites. $t$ test; *$P < 0.05$; $N = 15$ per group. **c** Confocal images of primary hippocampal neurons showing the effect of DHPG on PIP$_2$ levels in neurites. Bar, 50 μm. **d** Quantification of relative PIP$_2$ intensity in dendrites (upper) and axons (bottom) in control vs. DHPG conditions. $t$ test; **$P < 0.01$; ***$P < 0.001$; $N = 17$–19 per groups. **e–h** Representative traces of mEPSCs (**e**) and PPF (**g**), and quantification of mean values of mEPSC amplitude (**f**, left) and frequency (**f**, right) and PPF ratio (**h**) before (ctrl) and after DHPG treatment. $t$ test; **$P < 0.01$; $N = 8$ per group. **i** Confocal images of primary hippocampal neurons showing the effect of Aβ on PIP$_2$ levels in neurites in the presence of MTEP. Bar, 50 μm. **j** Quantification of relative PIP$_2$ intensity in dendrites (upper) and axons (bottom) in control vs. MTEP conditions. One-way ANOVA with post hoc Dunnett's test; $F_{(2,48)} = 0.23$ (upper); $F_{(2,48)} = 0.51$ (bottom); $P > 0.05$; $N = 16$–19 per group. **k, l** Representative traces (**k**) and the time course of the normalized amplitude (**l**) of SC-CA1 EPSCs in WT hippocampal slices before (BL) and after drug treatment (Veh, Aβ, Aβ + MTEP, or MTEP). **m** Quantification of relative amplitude of SC-CA1 EPSCs recorded in the last 1 min of drug treatment shown in (**l**). One-way ANOVA with post hoc Dunnett's test; $F_{(3,24)} = 14.1$; *$P < 0.05$; ***$P < 0.001$; $N = 5$–7 per group. **n–q** Representative traces of mEPSCs (**n**) and PPF (**p**), and quantification of mEPSC amplitude (**o**, left) and frequency (**o**, right) and PPF ratio (**q**) before (Veh) and after Aβ + MTEP treatment. $t$ test; **$P < 0.01$; $N = 7$–8 per group. Data are mean ± SEM. Source data are provided as a Source Data file

pyramidal neurons (Supplementary Fig. 8e, f), and PPF at the SC-CA1 synapse (Supplementary Fig. 8g, h) in WT animals. Furthermore, long-term treatment with MTEP (>3 h) greatly increased CA1 mEPSC frequency (Supplementary Fig. 11a, b) and SC-CA1 EPSC amplitude (Supplementary Fig. 11c, d) and reduced PPF at the SC-CA1 synapse (Supplementary Fig. 11e, f) in 6–7-month-old APP/PS1 mice, although this long-term maintenance of hippocampal slices in vitro caused a rundown in the baseline transmission. Taken together, these results imply that activation of presynaptic mGluR5 by Aβ oligomers contributes to PIP$_2$ depletion-associated reduction in Pr.

**Inhibiting presynaptic PIP$_2$ drop rescues Pr and memory in AD.** To establish fully that oligomeric Aβ-induced depletion of presynaptic PIP$_2$ underlies Pr reduction, selectively inhibiting Aβ-induced hydrolysis of presynaptic PIP$_2$ is essential. One way to control the PIP$_2$ level is to manipulate the responsiveness of Gαq-coupled GPCRs. The mammalian pho eighty-five requiring 3 (Efr3) proteins can control GPCR responsiveness[48]. Indeed, halving *Efr3a* in cultured astrocytes from *Efr3a*$^{+/−}$ mice resulted in a drastic decrease in DHPG-induced or Aβ-induced increase in intracellular Ca$^{2+}$ concentration ([Ca$^{2+}$]$_i$) that could be completely blocked by MTEP (Supplementary Fig. 12), indicating that knocking down *Efr3a* was an efficient way to suppress mGluR5 responsiveness. Importantly, we found that halving *Efr3a* copy number in APP/PS1 mice restored the decreased PIP$_2$ level in the AD mice (Supplementary Fig. 6). Oligomeric Aβ treatment was no longer effective in reducing total and neurite PIP$_2$ in cultured hippocampal neurons from *Efr3a*$^{+/−}$ mice (Fig. 7a–c). Furthermore, oligomeric Aβ$_{42}$ was less efficient to reduce SC-CA1 EPSCs in *Efr3a*$^{+/−}$ than in WT mice (Fig. 7d–f) and no longer altered mEPSC frequency in CA1 pyramidal neurons (Fig. 7g, h) and PPF at the SC-CA1 synapse in *Efr3a*$^{+/−}$ mice (Fig. 7i, j). Halving *Efr3a* copy number restored the decreased mEPSC frequency in CA1 pyramidal neurons and the upregulated PPF at the SC-CA1 synapse in APP/PS1 mice (Supplementary Fig. 13). By contrast, oligomeric Aβ$_{42}$ caused a more robust inhibition of mEPSC frequency in CA1 pyramidal neurons in Efr3a overexpression mice (Supplementary Fig. 14). Taken together, these results indicate that reducing *Efr3a* is an effective method to inhibit Aβ-induced PIP$_2$ depletion. Therefore, it is feasible to inhibit Aβ-induced PIP$_2$ hydrolysis region-specifically by creating conditional knockouts of *Efr3a* in the CA3 or CA1 area in mice.

We thus created conditional *Efr3a* knockouts in the CA3 and CA1 areas by crossing *Efr3a-loxP* mice to *Grik4-cre* and *CamKIIa-cre* mice, respectively (Fig. 8). In CA1-specific *Efr3a* conditional knockout (CA1-*Efr3a* cKO) mice 4–5 months of age (Fig. 8a) oligomeric Aβ$_{42}$ enhanced PPF at the SC-CA1 synapse

(Fig. 8b, c), implicating that deleting *Efr3a* at the postsynaptic site did not influence Aβ-induced Pr reduction. The minimal effect of deleting *Efr3a* in the CA1 area on Aβ-induced inhibition of Pr at the SC-CA1 synapse was further confirmed by directly examining the RRP size and Pr using the repetitive stimulation protocol (Fig. 8d, e). In CA3-specific *Efr3a* conditional knockout (CA3-*Efr3a* cKO) mice 4–5 months of age (Fig. 8f), however, oligomeric Aβ$_{42}$ was no longer effective in increasing PPF (Fig. 8g, h) and inhibiting Pr (Fig. 8i, j) at the SC-CA1 synapse, indicating that knocking out *Efr3a* at the presynaptic site rescued Aβ-induced Pr decrease. We then investigated whether selectively deleting *Efr3a* in the CA1 or CA3 area in APP/PS1 mice regulated Pr at the SC-CA1 synapse. In comparison to control mice, enhanced PPF (Fig. 8k, l) and decreased Pr (Fig. 8m, n) at the SC-CA1 synapse typical of 6–7-month-old APP/PS1 mice were still observed in age-matched APP/PS1 mice deleted for *Efr3a* in the CA1 area. In 6–7-month-old APP/PS1 mice deleted for *Efr3a* in the CA3 area, however, enhanced PPF and decreased Pr at the SC-CA1 synapse were restored to the control level (Fig. 8o–r). These results demonstrate that selectively deleting *Efr3a* at the presynaptic site of the SC-CA1 synapse effectively rescues the reduced Pr in APP/PS1 mice. Taken together, these data prove that inhibiting Aβ-induced presynaptic PIP$_2$ depletion efficaciously reduces Aβ-induced Pr reduction.

The SC-CA1 synapse plays an essential role in learning and memory. Therefore, it will be interesting to explore whether lowering Aβ-induced presynaptic PIP$_2$ depletion in the CA3 area improves the cognitive function in AD. To this end, we first investigated whether Aβ-induced presynaptic defect affected long-term potentiation (LTP) at the SC-CA1 synapse in WT and *Efr3a* conditional knockout mice (Fig. 9). A high-frequency train stimulation (3 bursts of 20 pulses at 100 Hz separated by 1.5 s) induced an apparent LTP of field excitatory postsynaptic potentials (fEPSPs) in WT mice (Fig. 9a–c), which had a robust presynaptic element as assessed by PPF (Fig. 9d–f) and was dramatically decreased in the presence of oligomeric Aβ$_{42}$ (Fig. 9a–c). Blocking mGluR5 with MTEP or deleting *Efr3a* in CA3 but not CA1 areas significantly ameliorated Aβ-induced LTP impairment (Fig. 9a–c), accompanying a decreased PPF following LTP induction (Fig. 9d–f). These results indicate that decreasing presynaptic GPCR responsiveness prevents oligomeric Aβ$_{42}$-induced LTP impairment.

We next studied if deleting *Efr3a* in the CA3 area ameliorated LTP impairment and improved cognitive function in APP/PS1 mice. As expected, the high-frequency train stimulation induced a dramatically decreased LTP in 6–7-month-old APP/PS1 mice in comparison to WT animals of the same age (Fig. 10a–c). Deleting *Efr3a* in the CA3 but not in the CA1 area significantly restored the diminished LTP in APP/PS1 mice (Fig. 10a–c). These data

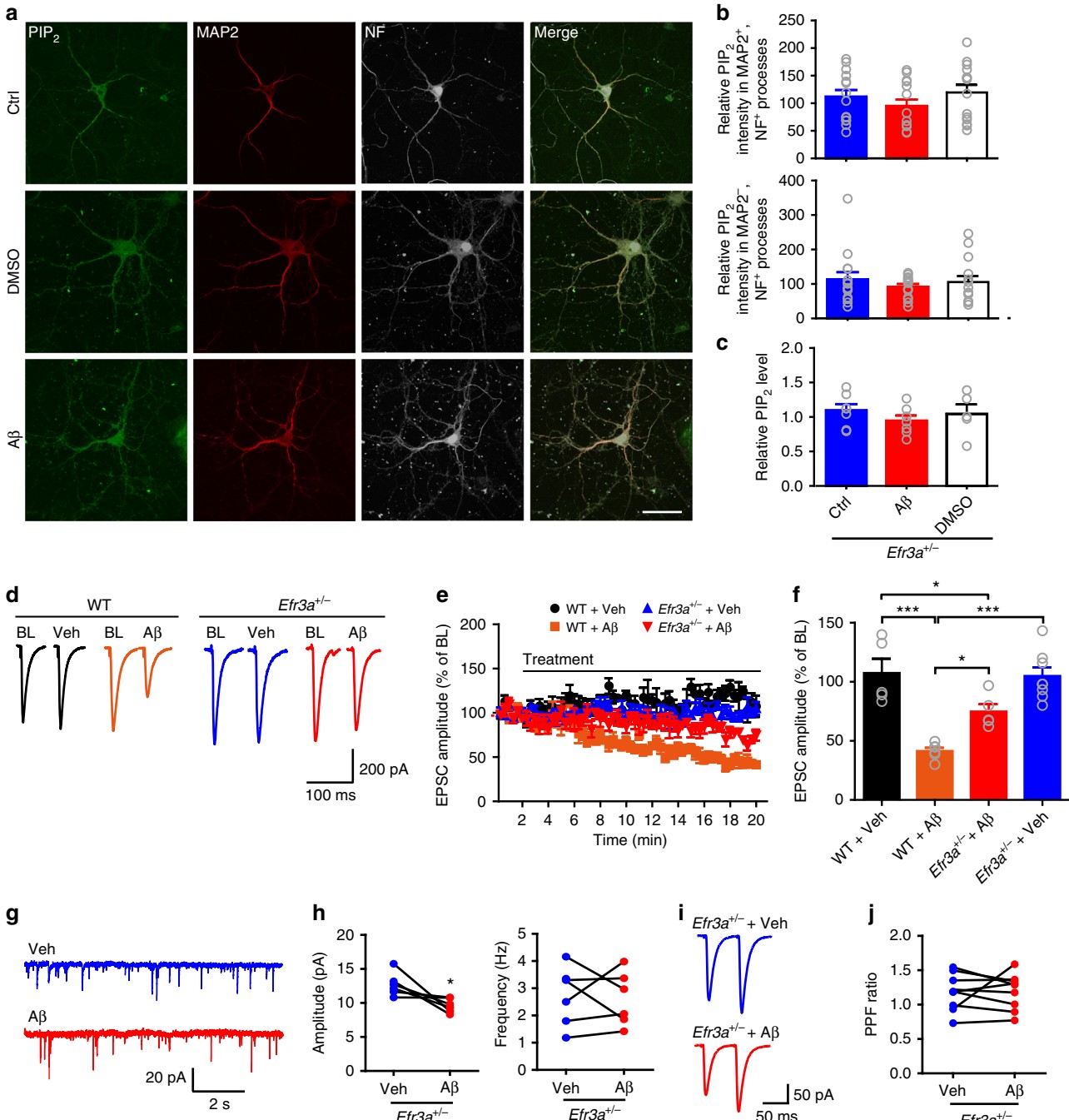

**Fig. 7** Oligomeric Aβ-induced PIP₂ reduction and presynaptic deficit are rescued by knocking down *Efr3a*. **a** Confocal images showing the effect of oligomeric Aβ on PIP₂ levels in neuronal processes in primary hippocampal neurons from *Efr3a*⁺/⁻ mice. Bar, 50 μm. **b** Quantification of relative PIP₂ intensity in dendrites (upper panel) and axons (bottom panel) showing knocking down *Efr3a* prevents Aβ-induced suppression of PIP₂ in neurites. One-way ANOVA with post hoc Dunnett's test; $F_{(2,39)} = 0.98$ (upper); $F_{(2,49)} = 0.44$ (bottom); $P > 0.05$; $N = 14$ per group. **c** Quantitative results showing knocking down *Efr3a* prevents Aβ-induced suppression of PIP₂ in cultured hippocampal neurons (measured by PIP₂ ELISA). One-way ANOVA with post hoc Dunnett's test; $F_{(2,16)} = 0.66$; $P > 0.05$; $N = 5$–7 per group. **d**, **e** Representative traces (**d**) and the time course of normalized amplitude (**e**) of SC-CA1 EPSCs in WT and *Efr3a*⁺/⁻ hippocampal slices before (BL) and after drug treatment (Veh or Aβ). **f** Bar graph representing the relative magnitude of EPSCs recorded in the last 1 min of drug treatment shown in **e**. One-way ANOVA with post hoc Dunnett's test; $F_{(3,23)} = 16.88$; *$P < 0.05$; ***$P < 0.001$; $N = 5$–8 per group. **g**, **h** Representative traces (**g**) and quantification (**h**) of mean values of the amplitude (left) and frequency (right) of mEPSCs in CA1 pyramidal neurons in *Efr3a*⁺/⁻ mice showing oligomeric Aβ treatment no longer inhibits mEPSC frequency. *t* test; *$P < 0.05$; $N = 6$ per group. **i**, **j** Representative traces (**i**) and quantification (**j**) of PPF of SC-CA1 EPSCs showing oligomeric Aβ does not alter PPF ratio in *Efr3a*⁺/⁻ mice. *t* test; $P > 0.05$; $N = 9$ per group. Data are mean ± SEM. Source data are provided as a Source Data file

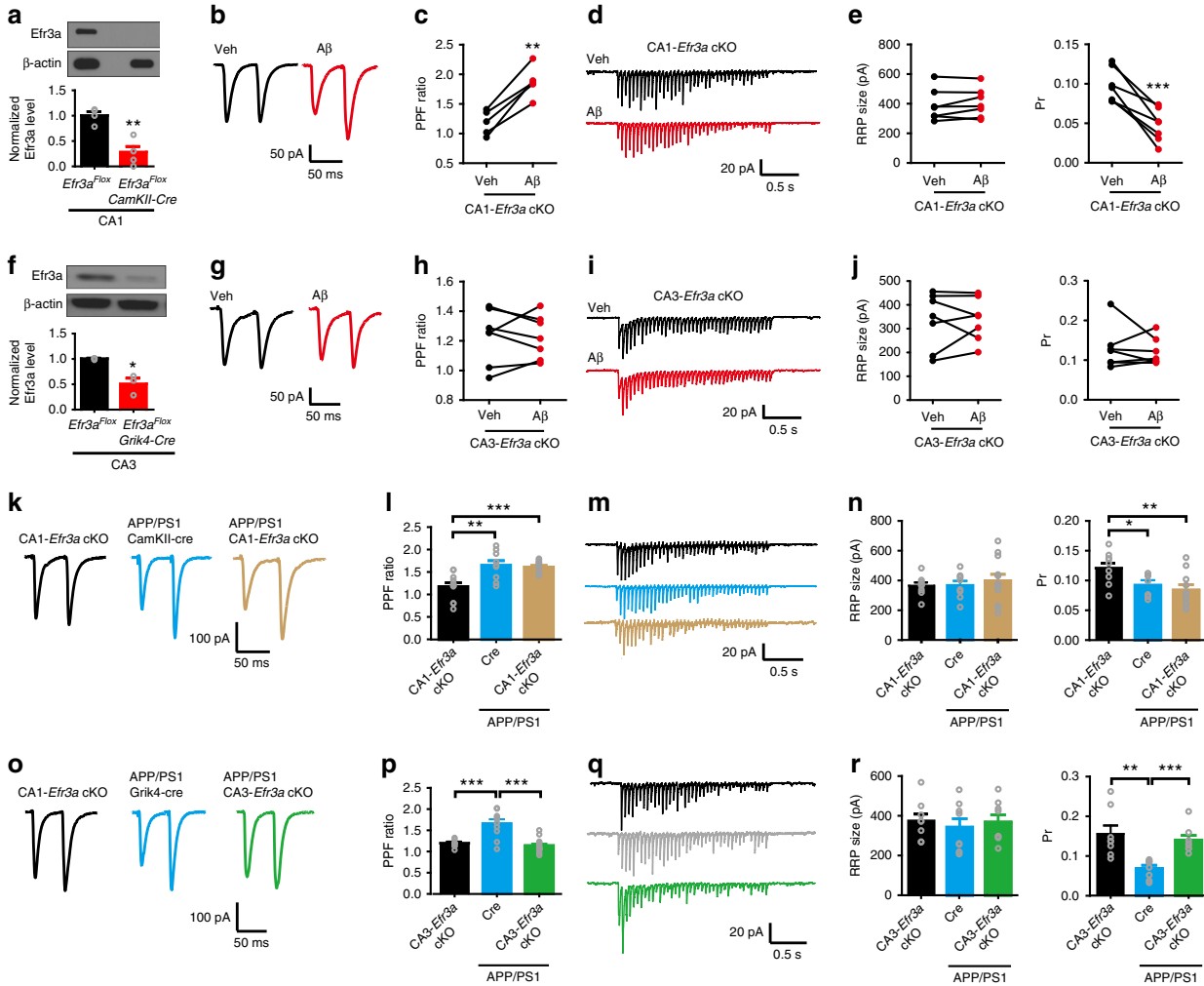

**Fig. 8** Selectively knocking out *Efr3a* in the CA3 area prevents oligomeric Aβ-induced inhibition of presynaptic release probability at the SC-CA1 synapse. **a–j** Representative traces of PPF (**b**, CA1-*Efr3a* cKO mice; **g**, CA3-*Efr3a* cKO mice) and 20 Hz train responses (**d**, CA1-*Efr3a* cKO mice; **i**, CA3-*Efr3a* cKO mice) and quantification of PPF ratio (**c**, CA1-*Efr3a* cKO mice; **h**, CA3-*Efr3a* cKO mice), RRP size (**e**, CA1-*Efr3a* cKO mice; **j**, CA3-*Efr3a* cKO mice; left), and Pr (**e**, CA1-*Efr3a* cKO mice; **j**, CA3-*Efr3a* cKO mice; right) at the SC-CA1 synapse showing oligomeric Aβ (400 nM) increases Pr in CA1-*Efr3a* cKO mice (immunoblotting and quantification of Efr3a in CA1 shown in **a**), whereas oligomeric Aβ no longer changes Pr in CA3-*Efr3a* cKO mice (immunoblotting and quantification of Efr3a in CA3 shown in **f**). *t* test; *$P < 0.05$; **$P < 0.01$; ***$P < 0.001$; $N = 3$–7 per group. **k–r** Representative traces of PPF (**k**, APP/PS1 CA1-*Efr3a* cKO and control mice; **o**, APP/PS1 CA3-*Efr3a* cKO and control mice) and 20 Hz train responses (**m**, APP/PS1 CA1-*Efr3a* cKO and control mice; **q**, APP/PS1 CA3-*Efr3a* cKO and control mice) and quantification of PPF ratio (**l**, APP/PS1 CA1-*Efr3a* cKO and control mice; **p**, APP/PS1 CA3-*Efr3a* cKO and mice), RRP size (**n**, APP/PS1 CA1-*Efr3a* cKO and control mice; **r**, APP/PS1 CA3-*Efr3a* cKO and control mice; left), and Pr (**n**, APP/PS1 CA1-*Efr3a* cKO and control mice; **r**, APP/PS1 CA3-*Efr3a* cKO and mice; right) at the SC-CA1 synapse showing selectively knocking out Efr3a in the CA3 area restores the decreased Pr in APP/PS1 mice. One-way ANOVA with post hoc Dunnett's test; $F_{(2,29)} = 11.38$ (**l**); $F_{(2,31)} = 20.66$ (**p**); $F_{(2,26)} = 0.28$ (**n**, left); $F_{(2,26)} = 5.99$ (**n**, right); $F_{(2,21)} = 0.986$ (**r**, left); $F_{(2,21)} = 0.19$ (**r**, right); *$P < 0.05$; **$P < 0.01$; ***$P < 0.001$; $N = 8$–15 per group. Data are mean ± SEM. Source data are provided as a Source Data file

indicate that selectively knocking out presynaptic *Efr3a* enhanced the SC-CA1 synapse plasticity in APP/PS1 mice, suggesting that decreasing the presynaptic GPCR sensitivity may improve the cognitive function of APP/PS1 mice. By comparing the performance of 6–7-month-old APP/PS1 mice in the Morris water maze (MWM) task to that of age-matched WT animals, we were able to reliably separate APP/PS1 mice from their WT controls in the escape latency and target time plots (Fig. 10d, e). Deleting *Efr3a* in the CA3 but not in the CA1 area in 6–7-month-old APP/PS1 mice significantly shortened the escape latency and increased the time spent in the target quadrant to the levels of WT mice in the MWM test (Fig. 10d, e). These results implicate that deleting *Efr3a* in the CA3 area improves spatial learning and memory in APP/PS1 mice.

## Discussion

Our work demonstrates a key presynaptic target of pathogenic Aβ in early AD and clarify Aβ-induced depletion of PIP$_2$ underlies Pr reduction at an excitatory synapse in the hippocampus. Aβ-induced activation of presynaptic mGluR5 depletes membrane PIP$_2$ in axons, which in turn disrupts neurotransmitter release. Notably, reducing Aβ-induced PIP$_2$ depletion in the CA3 area augments release Pr at the SC-CA1 synapse and enhances spatial learning and memory in APP/PS1 mice. As this Aβ-induced Pr reduction precedes synapse and neuronal loss, controlling the CA3 PIP$_2$ level may become an effective way of preventing AD progression.

We found that reduced mEPSC frequency in CA1 pyramidal neurons was a robust hallmark in 6–7-month-old APP/PS1 mice.

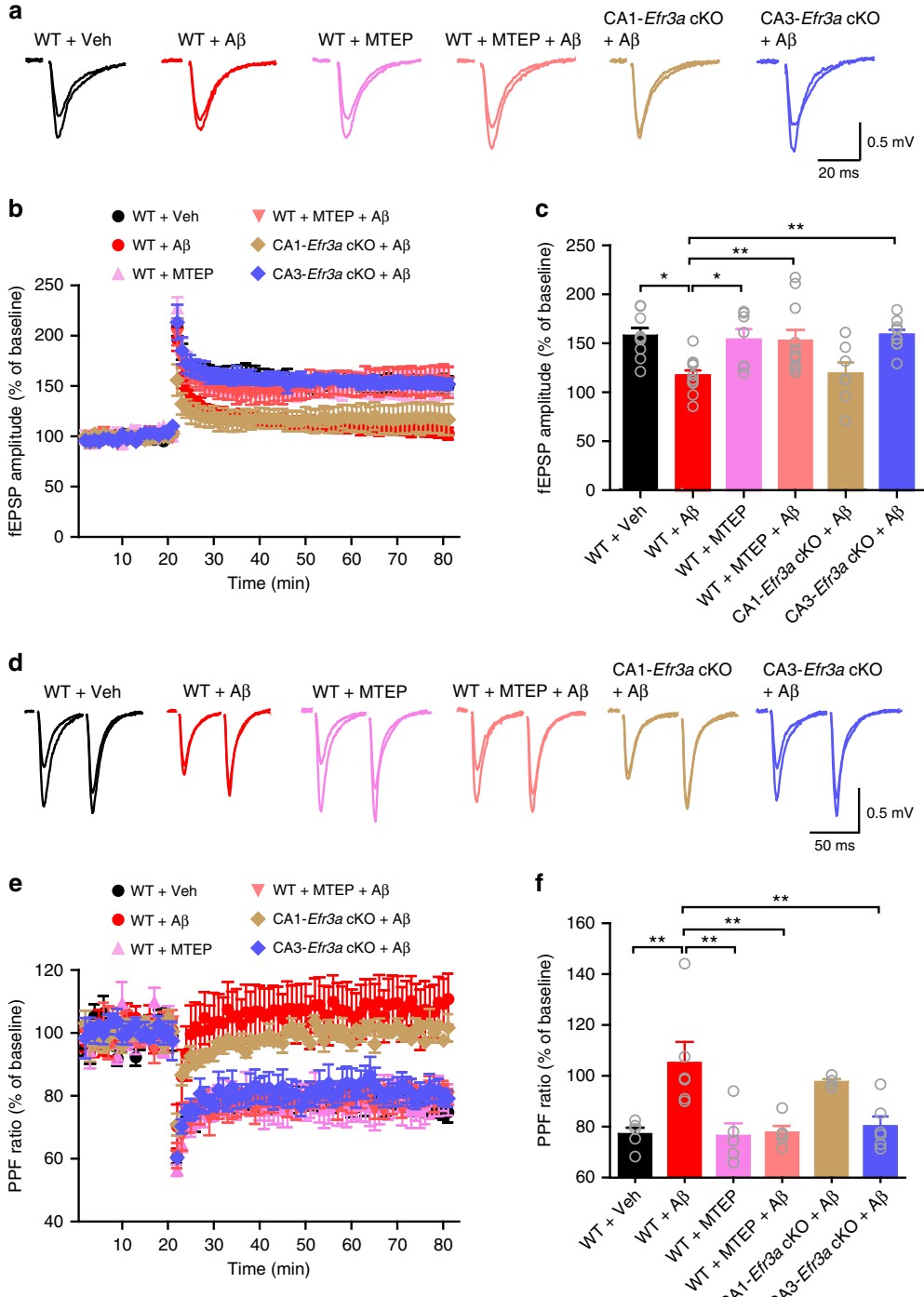

**Fig. 9** Blocking mGluR5 with MTEP or selectively knocking out *Efr3a* in the CA3 area ameliorates oligomeric Aβ-induced impairment of a presynaptically-expressed LTP at the SC-CA1 synapse. **a** Representative traces showing averaged fEPSPs recorded in CA1 area 10 min before and 50 min after high-frequency SC stimulation (superimposed) in WT slices treated with vehicle (WT + Veh), oligomeric Aβ (WT + Aβ), MTEP (WT + MTEP), or both MTEP and oligomeric Aβ (WT + MTEP + Aβ), or in oligomeric Aβ-treated hippocampal slices from mice selectively knocking out *Efr3a* in the CA1 (CA1-*Efr3a* cKO + Aβ) or CA3 (CA3-*Efr3a* cKO + Aβ) area. **b** The time course of the normalized amplitude of fEPSPs before and after an LTP induction protocol in conditions shown in **a**. **c** Bar graph representing mean LTP magnitude 50 min after LTP induction shown in **b**. One-way ANOVA with post hoc Dunnett's test; $F_{(5,48)} = 5.171$; *$P < 0.05$; **$P < 0.01$; $N = 7–11$ per group. **d** Representative traces of PPF of fEPSPs recorded in the CA1 area 10 min before and 50 min after high-frequency SC stimulation in the same conditions as in **a**. **e** The time course of the normalized PPF ratio before and after an LTP induction in the same conditions as in **d**. **f** Bar graph showing the relative PPF ratio before and after an LTP protocol. One-way ANOVA with post hoc Dunnett's test; $F_{(5,25)} = 6.361$; **$P < 0.01$; $N = 4–6$ per groups. Data are mean ± SEM. Source data are provided as a Source Data file

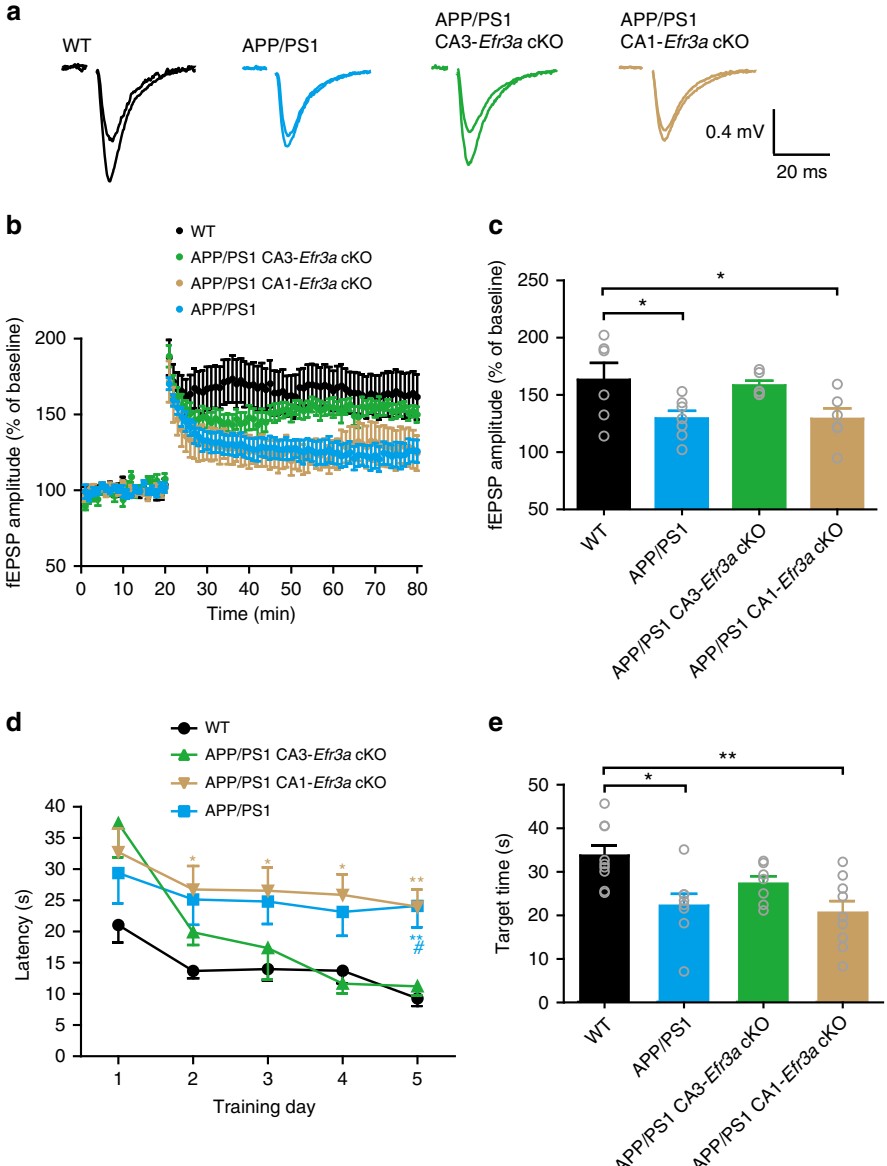

**Fig. 10** Selectively knocking out *Efr3a* in the CA3 area ameliorates impairment in a presynaptically-expressed LTP at the SC-CA1 synapse and spatial learning and memory deficits in APP/PS1 mice. **a** Representative traces of averaged fEPSPs recorded in the CA1 area 10 min before and 50 min after high-frequency SC stimulation (superimposed) in WT, APP/PS1, APP/PS1 CA1-*Efr3a* cKO, and APP/PS1 CA3-*Efr3a* cKO mice. **b** Summary time course of the normalized amplitude of fEPSPs recorded from WT, APP/PS1, APP/PS1 CA1-*Efr3a* cKO, and APP/PS1 CA3-*Efr3a* cKO mice during an LTP protocol. **c** Bar graph representing mean LTP magnitude recorded 50 min after LTP induction shown in **b**. One-way ANOVA with post hoc Dunnett's test; $F_{(3,21)} = 3.87$; *$P < 0.05$; $N = 6$–7 per group. **d** Quantification of the escape latency in each session of the hidden-platform test for WT, APP/PS1, APP/PS1 CA1-*Efr3a* cKO, and APP/PS1 CA3-*Efr3a* cKO mice. Two-way ANOVA with post hoc Bonferroni test; animal, $F_{(3,145)} = 15.78$, $P < 0.001$; training session, $F_{(4,145)} = 9.51$, $P < 0.001$; #$P < 0.05$ (APP/PS1 CA3-*Efr3a* cKO vs. APP/PS1); *$P < 0.05$; **$P < 0.01$ (blue *, WT vs. APP/PS1; brown *, WT vs. APP/PS1 CA1-*Efr3a* cKO); $N = 7$–9 per group. **e** Bar graph showing the mean target quadrant searching time in the probe test for WT, APP/PS1, APP/PS1 CA1-*Efr3a* cKO, and APP/PS1 CA3-*Efr3a* cKO mice. One-way ANOVA with post hoc Dunnett's test; $F_{(3,29)} = 6.13$; *$P < 0.05$; **$P < 0.01$; $N = 7$–9 per groups. Data are mean ± SEM. Source data are provided as a Source Data file

CA1 pyramidal neurons mainly receive excitatory inputs from CA3 pyramidal neurons and layer III pyramidal neurons in the entorhinal cortex. Thus, Aβ-induced changes in mEPSC frequency in CA1 pyramidal neurons may be attributed to presynaptic alterations in both glutamatergic terminals forming synapses on CA1 pyramidal neurons. Although we are not able to rule out the involvement of the excitatory inputs from the entorhinal cortex completely, the decrease in mEPSC frequency may primary be associated with Aβ-induced suppression of glutamate release from the SC. Reducing the Aβ sensitivity in

CA3 pyramidal neurons not only increased the amplitude of SC-CA1 EPSCs but also restored the diminished mEPSC frequency in CA1 pyramidal neurons (Supplementary Fig. 15). Because both areas CA3 and CA1 are critical in encoding memory sequences, our results implicate that reduced glutamate release at the SC-CA1 synapse represents a crucial early event leading to cognitive decline in AD.

Our results showed that 400 nM oligomeric Aβ₄₂ reduced glutamate release from the SC in WT mice to a level similar to that in 6–7-month-old APP/PS1 and 4-month-old 5xFAD mice.

In another APP/PS1 transgenic line APP$_{Swe}$/PS1$_{M146L}$, however, there is no reduction in Pr[49]. One possible explanation is that the effective concentration of oligomeric Aβ has to rise significantly at the target synapse to suppress Pr in early AD. Although in no way could we fully determine the effective concentration of locally distributed soluble Aβ, the concentration of the most toxic oligomeric Aβ$_{42}$ has to rise to hundreds of nanomolar level to inhibit Pr. A moderate increase of Aβ to the picomolar level[19], or even 25-fold increase of Aβ$_{42}$ found in 4-month-old APP/PS1 mice[50], usually cause an increase in Pr at the SC-CA1 synapse, leading to enhanced basal synaptic transmission[19] that we also observed with 20 nM oligomeric Aβ$_{42}$. A moderate increase in intracellular Aβ oligomers can also increase evoked responses by Ca$^{2+}$-dependent insertion of GluA1 subunits[51]. A further increase in Aβ oligomers to the low micromolar level, however, induces mostly postsynaptic depression and loss of dendritic spines[8,9]. Thus, although we are not in a position to define what the pathological level of oligomeric Aβ is[8], we believe the early presynaptic deficit in AD requires elevation of soluble Aβ at least in the high nanomolar range. On the other hand, the facilitating effect of small increases in Aβ within a physiological range on presynaptic neurotransmitter release represents a physiological function of oligomeric Aβ[19].

The probability of neurotransmitter release is tightly associated with Ca$^{2+}$-dependent synaptic vesicle fusion in the presynaptic terminal. In contrast to enhancing Pr by physiological range of Aβ via increasing presynaptic Ca$^{2+}$, our results implicate that high nanomolar Aβ mainly suppresses Pr via reducing PIP$_2$-dependent presynaptic vesicle fusion. In cultured neurons, 200 nM Aβ can also enhance Pr[42], suggesting that Aβ may interfere with presynaptic function in a PIP$_2$-independent manner. However, Aβ-induced Pr suppression may not be apparent in other central synapses (e.g., the recurrent inputs to CA3 pyramidal neurons)[52]. It will be interesting to investigate if these synapses lack such a mechanism or a compensatory process engages following Aβ accumulation in future studies. Reduced Pr exists in other AD models. Conditionally knocking out presenilins 1 and 2 in CA3 pyramidal neurons significantly inhibited Ca$^{2+}$ release from ryanodine receptors, thus reducing Pr at the SC-CA1 synapse[53]. Our results established that maintaining the membrane PIP$_2$ at a relatively high level in axons is critical in preserving a proper Pr (Fig. 4). A low PIP$_2$ level in the brain is a hallmark for both aging and AD animals[38,54]. However, in normal aging mice Pr is not altered. This could be due to a greater PIP$_2$ hydrolysis initiated by oligomeric Aβ in AD than that in normal aging animals with a low PLC expression level[55]. PIP$_2$ interacts with synaptotagmin 1 and Munc13–2[46,56] to control exocytosis. Future studies are required to address the detailed mechanism underlying PIP$_2$-dependent Pr regulation in AD.

One of the major findings of our study is that inhibiting mGluR5 rescued the presynaptic deficit in early AD. Genetically deleting[36] or pharmacologically inhibiting[35,37] mGluR5 has been shown to significantly improve cognitive impairment in AD mice, as mGluR5 may function as an Aβ receptor or co-receptor with PrP$^C$ in both APP overexpression and knock-in mouse models of AD[32–34]. Our results clearly show that presynaptic mGluR5-mediated PIP$_2$ hydrolysis requires PrP$^C$ as well. Although group I mGluRs are expressed in both presynaptic and postsynaptic loci[57], mGluR5 density is higher in dendrites than in axons (Fig. 6a, b). However, high nanomolar Aβ oligomers deplete PIP$_2$ to a greater level in axons than in dendrites. Although we do not know exactly what causes this differentially regulation of PIP$_2$ in neurites, we nevertheless speculate that this may be due to a combination of enhanced mGluR5 receptor responsiveness and downstream process efficacy in axons.

Functionally, we believe presynaptic mGluR5 may be a main target of soluble Aβ in early AD based on the following observations, although postsynaptic mGluR5 has been shown to be involved in Aβ-induced suppression of LTP and enhancement of LTD[29,30,58]. First, blocking mGluR5 did not restore nanomolar oligomeric Aβ$_{42}$-induced suppression of mEPSC amplitude, implicating that activation of postsynaptic mGluR5 is not involved in the oligomeric Aβ-induced mild postsynaptic defect. Although the detailed mechanisms require further examination, removal of surface AMPA and NMDA receptors may account for this oligomeric Aβ-induced reduction of mEPSC amplitude[58,59]. On the other hand, micromolar oligomeric Aβ induce mature spine loss via mGluR5-mediated downregulation of CaMKII activity in the postsynaptic site in APP knock-in mouse model of AD[60]. Second, selectively decreasing the sensitivity of presynaptic GPCRs ameliorated oligomeric Aβ-induced suppression of Pr and a form of LTP with a notable presynaptic element, thus ruling out indirect presynaptic effects due to activation of postsynaptic mGluR5. A presynaptically-expressed SC-CA1 LTP can be induced by 200 Hz[61,62] or multiple trains of 100 Hz[63,64] tetanus and may require postsynaptically-activated Ca$^{2+}$ influx through L-type calcium channels and NMDA receptors[61,62]. Although we do not know the postsynaptic mechanism underlying this presynaptically-expressed LTP, our results establish that Aβ oligomers suppress this LTP through presynaptic mGluR5.

In the current study, we controlled membrane PIP$_2$ hydrolysis by manipulating the Efr3a level. It has long been known that Efr3a functions as an adaptor in the type IIIα phosphatidylinositol 4-kinase (PI4KIIIα) complex[65,66]. Although PI4KIIIα is responsible for generating phosphatidylinositol 4-phosphate (PI4P), the precursor of PIP$_2$, reducing Efr3a or PI4KIIIα induces minimal changes in the membrane PIP$_2$ level[65,66]. Knocking out the type 1γ phosphatidylinositol phosphate kinase, the main enzyme responsible for PIP$_2$ synthesis at synapses, however, significantly depletes membrane PIP$_2$, leading to decreased transmitter release[67]. Efr3a may also control the responsiveness of other GPCRs that are putative Aβ receptors such as the type-1 angiotensin II receptor[48]. Thus, controlling the expression level of Efr3a may regulate a series of GPCRs that are targets of Aβ, though we believe mGluR5 is the main target of pathological level of Aβ. Nevertheless, antagonizing Efr3a[68] may provide a more robust way to treat AD rather than targeting only one type of GPCR.

## Methods

**Animals.** All procedures were carried out in accordance with the National Institutes of Health Guidelines for the Care and Use of Laboratory Animals and were approved by the Animal Advisory Committee at Zhejiang University. B6, APP/PS1 double-transgenic, 5xFAD, M146V, Grik4-cre, and Camk2a-creERT2 mice were purchased from The Jackson Laboratory (Bar Harbor, ME). *Efr3a* double-flox (*Efr3a$^{f/f}$*), *Efr3a$^{+/−}$* heterozygotes, *Efr3a$^{f/f}$-Grik4-cre*, *Efr3a$^{f/f}$-Camk2a-creERT2*, APP/PS1-*Efr3a$^{f/f}$-Grik4-cre*, APP/PS1-*Grik4-cre*, APP/PS1-*Efr3a$^{f/f}$-Camk2a-creERT2*, and APP/PS1-*Camk2a-creERT2* animals were obtained by heterozygous mating. Tamoxifen was intraperitoneally (i.p.) injected once a day for 5 consecutive days at a dose of 100 mg kg$^{−1}$ to induce cre recombinase expression in the creER lines. For behavioral experiments, only male mice were used. The mouse genotypes were identified by PCR using genomic DNA from mouse tails and embryo tissues. Primers and detailed protocols are available in the Supplementary Table 1 and Supplementary Methods. LY-411575 was used to inhibit γ-secretase. The protocols of LY-411575 preparation and treatment are detailed in Supplementary Table 2 and Supplementary Methods.

**Antibodies and drugs.** The following commercially available antibodies were used: rabbit anti-amyloid precursor protein, C-terminal fragments (anti-CTFs), purified mouse anti-β-Amyloid, 1–16 (6E10), rabbit anti-Efr3a, mouse anti-PLCβ1, D-8, mouse anti-PLCβ4, A-8, mouse anti-β-actin, and HRP-conjugated secondary antibodies were used in Western blotting; mouse anti-PIP$_2$ antibody, chicken anti-neurofilament-L (anti-NF), rabbit anti-MAP2 antibody, rabbit anti-mGluR5 (extracellular), mouse anti-MAP2, purified anti-CD230 (Prion) antibody, and Alexa Fluor-conjugated secondary antibodies (488 donkey anti-rabbit, 546 donkey

anti-mouse, 546 donkey anti-rabbit, 405 goat anti-rabbit IgG H&L, 488 goat anti-mouse IgM mu chain and 647 goat anti-chicken IgY H&L) were used in immunocytochemistry. The antibody information is detailed in Supplementary Methods.

Tamoxifen was dissolved in 100% ethanol; PI(4,5)P$_2$ diC8 was dissolved in the electrode solution; DHPG, MTEP, and U73122 were dissolved in the bath solution or culture medium. The final concentrations of DMSO did not exceed 0.1% throughout the study. The protocol of oligomeric A$\beta_{42}$ preparation and drug information are detailed in Supplementary Table 2 and Supplementary Methods.

**Golgi staining**. Golgi staining was carried out using an FD Rapid GolgiStain Kit according to the manufacturer's instructions (see details in Supplementary Methods).

**Slice recording**. Briefly, mice (4- or 6–7-month-old) were anesthetized with isoflurane and decapitated, and transverse slices of hippocampus (300 μm for whole-cell recording or 350 μm for fEPSP recording) were cut with a tissue slicer (VT 1200S, Leica) in oxygenated ACSF (for whole-cell recording) or in oxygenated cutting solutions (for fEPSP recording). Whole-cell recordings were performed on CA1 pyramidal neurons[69,70]. mEPSCs signals were recorded at −70 mV in ACSF containing 0.5 μM tetrodotoxin (TTX) and 10 μM bicuculline. Evoked EPSCs were elicited in the presence of 10 μM bicuculline using a bipolar stimulating electrode placed in stratum radiatum 300 μm away from the recording site. PPF experiments were carried out by delivering a pair of stimuli with an interval of 50 ms. To estimate the RRP size and release Pr, a repeated 20 Hz train stimulation protocol was used to evoke 40 EPSCs. The RRP size was calculated by linear interpolating the linear portion of the cumulative EPSC amplitude plot to virtual stimulus 0. The release Pr was calculated as the mean amplitude of the 1st EPSC during the repeated train stimulations divided by the RRP size. fEPSPs were elicited by stimulating the SC and recording with a borosilicate glass electrode filled with ACSF placed in CA1 stratum radiatum. LTP was induced by 3 bursts of 20 pulses at 100 Hz separated by 1.5 s. Detailed protocols are available in Supplementary Methods.

**Cell culture**. Primary hippocampal neuron cultures were prepared from embryonic day 18 (E18) mice[69]. Briefly, embryos were removed from maternal mice anesthetized with isoflurane and euthanized by decapitation. Hippocampi were dissected in HBSS, followed by a digestion with 0.25% w/v trypsin. Neurons were centrifuged (1000 × g for 5 min) and resuspended in neurobasal medium containing 2% B27 serum-free supplement, 1% v/v penicillin/streptomycin (P/S), 0.5 mM glutamine, and 10 μM glutamate. Dissociated cells were then plated with appropriate densities in culture plates or dishes pre-coated with PDL. Cultures were kept at 37 °C in a 5% v/v CO$_2$ humidified incubator. Thereafter, one third to half of the medium was replaced twice a week (see details in Supplementary Methods).

Individually inhabited hippocampal neurons were grown on collagen-PDL islands. Briefly, 6.5 mm Transwell® inserts in 24 well plates were coated with PDL (12 h before culture), and coverslips were sprayed with island substrate solution containing 1 mg ml$^{-1}$ PDL and 3 mg ml$^{-1}$ rat tail collagen (3 h before culture). Dissociated cells were then plated at a density of 2000 cells per cm$^2$ onto coverslips in 24-well plates (for micro-island culture) or at a density of 50,000 cells per cm$^2$ in Transwell® inserts in 24-well plates (as high density neuronal feeder layer). After an adherence time of 4 h, the transwell inserts with neurons (high density) were placed into 24-well plates with neurons on coverslips (low density). Procedures for maintaining cultured islands were similar to those for primary hippocampal neuron cultures (see details in Supplementary Methods).

Astrocyte cultures were prepared from 0 to 1-day-old (P0–1) mice[69]. Cortices were dissected from 0 to 1-day-old mice and digested with 0.25% w/v trypsin in DMEM. Cells were allowed to grow for at least 7 days at 37 °C with 5% CO$_2$, and a complete medium change was performed every other day. At confluence after DIV8-10, cultures were shaken, and then incubated with 20 μM cytosine-1-β-D-arabinofuranosid (see details in Supplementary Methods).

**FM1-43 loading and synaptic vesicle detection**. Cultured neurons (DIV14) were transferred into a standard bath solution with 10 μM DNQX and 40 μM D-AP5. Neurons were then incubated with 5 μM FM1-43 in a hyperkalemic bath solution for 90 s. FM1-43 was then washout followed by adding ADVASEP-7 to reduce background fluorescence. Images were taken by a confocal laser-scanning microscope (Nikon A1). FM1-43-loaded vesicles were viewed through a 40X oil-immersion objective and images were acquired at a resolution of 1024 × 1024 pixel at RT (see details in Supplementary Methods).

**Western blotting**. Hippocampi were obtained and homogenized using a chilled Vibrahomogenizer (Vibra cell, SONICS) in 2 ml of RIPA buffer. The lysate was then centrifuged, and the supernatant collected for Western blot analysis. Proteins were separated on SDS-PAGE under denaturing conditions (for Efr3a, 10–15% Mini-PROTEAN TGX Gels; for Aβ and CTFs, 16.5% Tris-Tricine Gels) and transferred to polyvinylidene fluoride (PVDF) microporous membrane (Millipore). The membranes were then blocked with 5% skim milk-TBS (for Aβ) or 0.35% gelatin-TBST (for other proteins), and incubated with the primary antibodies followed by HRP-conjugated secondary antibodies. Protein bands were then

visualized using the ECL western blotting detection substrate and analyzed with ImageJ software. Detailed protocols are available in Supplementary Methods.

**Cultured neuron recording**. Single-cell micro-island neuron cultures at DIV14 were used for recording. Neurons were voltage clamped at −70 mV with a Heka EPC 10 amplifier and mEACs were recorded at 32 °C in bath solution containing 0.5 μM TTX and 10 μM bicuculline. Individual events were counted and analyzed with MiniAnalysis software (see details in Supplementary Methods).

**Lipid strip assay**. PIP Strips™ membranes were used for the anti-PIP$_2$ antibody specificity measurement according to the manufacturer's instructions. Detailed protocols are available in Supplementary Methods.

**ELISA PIP$_2$ assay**. Mass ELISA Kit K-4500 from Echelon Biosciences was used to determine PIP$_2$ levels in hippocampi from WT and APP/PS1 mice (6–7-month-old), and in primary cultured hippocampal neurons from WT and Efr3a$^{+/-}$ mice according to the manufacturer's instructions[55]. Detailed protocols are available in Supplementary Methods.

**Immunocytochemistry**. Immunofluorescence staining was carried out in cultured neurons at DIV14[69]. Briefly, neurons were fixed and permeabilized followed by incubation with primary antibodies and the appropriate secondary antibodies. For co-staining with mGluR5, neurons were permeabilized before the secondary blocking, and the primary antibodies were used. Fluorescent images were acquired through a 60X oil-immersion objective using a Nikon A1 confocal laser-scanning microscope. Neuronal images were analyzed using Meta-Morph with customized filter sets. Detailed protocols are available in Supplementary Methods.

**Lentivirus-shRNA infection**. Cultured hippocampal neurons at DIV7 were infected with lentivirus carrying DsRed-PLCβ1-shRNA (PLCβ1-shRNA) or DsRed-PLCβ4-shRNA (PLCβ4-shRNA) to knock down PLCβ1 or PLCβ4. Lentivirus expressing DsRed-scramble-shRNA was used as control. Neurons were then treated with Aβ or DMSO, and subsequently used for Western blotting (interference efficiency detection) or immunofluorescence staining. Detailed protocols are available in Supplementary Methods.

**Ca$^{2+}$ imaging**. Changes in [Ca$^{2+}$]$_i$ were measured in astrocytes (WT and Efr3a$^{+/-}$) using the calcium-sensitive fluorescent dye Fluo-4. The astrocytes were washed with Krebs buffer and incubated with 4 μM Fluo-4 in Krebs buffer. Fluo-4 loaded astrocytes were excited at 488 nm and fluorescence emission was detected at 525 nm. The images [(baseline data ($F_0$) and treated data ($F$)] were taken through a 60X oil-immersion objective by a Nikon A1 confocal laser-scanning microscope. The relative Fluo-4 fluorescent signals expressed in arbitrary units ($F/F_0$) were analyzed for individual cells using MetaMorph with a fixed set of parameters (see details in Supplementary Methods).

**MWM test**. The MWM tests were performed in a circular tank filled with opaque water at 25 °C. Twenty-four hours before the acquisition test, a visible platform task was tested in each quadrant of the tank. In the hidden platform acquisition test, mice could swim freely to search for the escape platform within 60 s. The time taken to reach the platform was recorded as the escape latency. Mice were allowed to stay on the platform for 10 s after the hidden platform was found. The same animal was then released from a new insertion point 4 min after the previous trial. The experiment was repeated four times per mouse each day for 5 days. Twenty-four hours after the hidden platform acquisition test, probe trials were conducted by removing the platform. The numbers of entries into the area where the original platform was located and crossings over the original platform were recorded. The data were analyzed by the WaterMaze Software (Actimetrics, INC.). Detailed protocols are available in Supplementary Methods.

**Statistics and reproducibility**. GraphPad Prism (Version 5.01, Graph-Pad Software Inc.) was used for data display and statistical analysis. We did not predetermine sample sizes. We used Kolmogorov-Smirnov normality test to determine most data are Gaussian distributed. Significance is reported as $P < 0.05$, and data were expressed as mean ± SEM. Two-tailed Student $t$ test, one-way ANOVA followed by a post hoc multiple comparison analysis based on the Dunnett's Method, two-way ANOVA followed by a post hoc Bonferroni test, or two-way RM ANOVA followed by a post hoc Bonferroni test were used to determine significant levels between treatments and controls. Distributions of mEPSC amplitudes and interevent intervals were compared using Kolmogorov-Smirnov test.

## Data availability

All data generated or analyzed during this study are included in this published article and its supplementary information files. The source data underlying Figs. 1–10 and Supplementary Figs 1–15 are provided as two Source Data files.

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

## Acknowledgements

This work was supported by the Major Research Program from the State Ministry of Science and Technology of China (2013CB530902), the National Natural Science Foundation of China grants (81770839, 81571125, 81571088, 91132712, 81300979, 81821091, 81521062, and 81221003), the 111 project, and the Non-profit Central Research Institute Fund of Chinese Academy of Medical Sciences (2017PT31038 and 2018PT31041). We thank the excellent technical assistant of the Imaging Facility at Zhejiang University School of Medicine.

## Author contributions

Y.-D.Z., Yi S. and F.-D.H. designed the study. Y.-D.Z., Yi S., Y.H. and M.W. wrote the paper. Yi S., Y.H., M.W., Y.W., X.M. and W.L. analyzed the data. Y.H. and X.M. performed electrophysiology experiments. Y.H. performed synaptic vesicle detection and Ca$^{2+}$ imaging experiments. M.W. and Y.W. did behavioral studies. M.W., H.Q., Y.W., J.C. and W.L. did immunostaining and biochemical studies. F.-D.H. engineered *Efr3a* transgenic mice. B.S., J.R., Z.C. and Ye S. contributed intellectually to the manuscript.

## Additional information

**Competing interests:** F.-D.H. has shareholding of a company possessing part of the intellectual property raised in this study. The remaining authors declare no competing interests.

**Journal Peer Review Information**: *Nature Communications* thanks the anonymous reviewer(s) for their contribution to the peer review of this work. Peer reviewer reports are available.

