## [Peer Review File · Nature Communications]

Reviewers' comments:

Reviewer #1 (Remarks to the Author):

Many studies have established a role for soluble oligomers of amyloid-beta 42 (Ab42) in the disruption of synaptic transmission and plasticity in vitro and in vivo. A dominant hypothesis in the field is that this phenomenon may contribute to the onset of cognitive deficits in the early stages of Alzheimer's disease (AD). The bulk of those studies have focused on the negative impact of Ab42 oligomers on postsynaptic processes, including perturbation of glutamate receptor/channel trafficking and dendritic spine dynamics. This study by He et al focuses on the presynaptic impact of Ab42 oligomers through a combination of in vitro and in vivo experiments (using both pharmacology and mouse genetics), establishing a role for metabotropic glutamate receptor 5 (mGluR5) and downstream signaling through phospholipase C (PLC) and phosphatidylinositol-4,5-bisphosphate (PIP2) hydrolysis in decreasing the release probability at hippocampal synapses.

Overall, this is an interesting study that expands on previous studies from several labs indicating that PIP2 hydrolysis is a pathophysiologically relevant consequence of Ab42 oligomer increase in the brain of AD mouse models as well as in cultured hippocampal neurons. Like the other studies, the manuscript by He et al does not provide much better understanding of the molecular mechanisms occurring downstream of PIP2 hydrolysis, and specifically which PIP2 (or DAG) effectors mediate the synapse impairing actions of Ab42 oligomers. However, this study does a great job establishing a role for mGluR5 in these actions, supporting its main conclusions with elegant electrophysiologic analyses and rather compelling results. There are however several aspects that need to be addressed to make this manuscript worthy of publication in Nature Communications, in this referee's view. These are as follows:

1. Having established that PIP2 hydrolysis occurs downstream of Ab42 via mGluR5, it is important to determine which PLC isoforms mediate this effect presynaptically. The broad PLC inhibitor is used in this study as well as in other studies that have examined the impact of Ab42 on PIP2 metabolism, but short of elucidating the effect of PIP2 hydrolysis on molecular pathways controlling the release probability, the authors could at least determine which PLC isoform is responsible for PIP2 hydrolysis. Granted that there are multiple PLC isoforms, it appears as though PLCbeta1 and beta4 may be good candidates. Simple shRNA experiments should allow the authors to determine whether they are the ones involved.
2. This referee is not a big fan of anti-PIP2 immunostainings given the notorious artifacts reported by many investigators in the field. The in vitro studies examining PIP2 with antibodies should be complemented with studies where PH domains (fused to fluorescent proteins) are used to assess PIP2 levels microscopically. Also, they could use the same PIP2 ELISA they have employed to measure PIP2 levels in APP-PSEN1 mouse brains.

Other comments:

3. The PIP2 depletion induced by Ab42 oligomers was already shown to be PLC-dependent in reference 40. Rather than describing their findings as novel on line 159, the authors should simply state that they confirmed previously published results.
4. The introduction focuses a lot on the physiological role of Ab42 at synapses, but the results presented in the manuscript are more related to its pathophysiological role. This referee recommends that the authors should shorten the former section and expand the latter, perhaps focusing more on the known effects of Abeta oligomers (and apoE4) on PIP2 metabolism and how levels of PIP2

phosphatase synaptojanin modulate these effects.

5. The reference "Deleterious effects of soluble amyloid- β oligomers on multiple steps of synaptic vesicle trafficking." By Park et al. *Neurobiol Dis.* 2013, 55:129-39. PMID: 23523634. Should be cited and discussed, since it focuses on the presynaptic impact of Ab42 oligomers.

Reviewer #2 (Remarks to the Author):

Yang He et al explore mechanisms of synaptic dysfunction in models of AD. They report that slices from APP/PS1 transgenic at 6-7 months of age show evidence for impaired presynaptic function with decreased release probability (Pr) and decreased miniature EPP frequency. At this stage little or no change in Golgi-impregnated synaptic spines was apparent. Acute application of A β oligomer induced the same phenomenon in WT slice. The authors provide data linking this Pr decrement to activation of mGluR5 and depletion of PIP2. They utilize knockout of Efr3a as a means to prevent PIP2 depletion. They examine the functional consequence of these observations by assessing LTP in CA3-CA1 pathway and show rescue with Efr3a knockout in CA3. The deletion also rescues learning but not memory in the water maze.

These data are intriguing and contribute to the field by exploring presynaptic actions of A β oligomers to disrupt synaptic function. The hypothesis of an early presynaptic deficit mediated via mGluR5 and PIP2 would add a new dimension to mechanistic understanding of synaptic function in AD. Most work has previously described synapse loss with both pre and post elements gone, and mechanistic investigations have documented post-synaptic changes in a range of AD models. There are several aspects of the work that require further data to clarify and relate to previous work.

1. The authors emphasize that the Pr presynaptic change is seen early in APP/PS1 mice prior to synapse loss and most previous studies showing any symptoms in this line. The authors must examine earlier ages to show whether this is some developmental issue in this transgenic line or whether development is normal at 2 or 3 months age and as A β starts to accumulate the presynaptic phenotype occurs in a disease-related process.

2. The mouse experiments utilize one AD transgenic line. Anyone line may produce idiosyncratic results, so verifying the basic observation of an early presynaptic Pr issue in a second line would greatly strengthen the paper. A knock-in line would be best to study.

3. The authors report altered PPF in the APP/PS1 slices. This has been studied previously and others have NOT observed PPF changes (to mention just two: *J Alzheimers Dis.* 2018 61:195-208, *Cell Death Dis.* 2015 6:1676). This should be reviewed, and any potential cause discussed. The authors focus on presynaptic changes does not match well with previous work.

4. The biochemical measure of PIP2 loss is in the Supplement only. It would be advantageous to show in main Figure.

5. The pharmacology implicates mGluR5 and downstream signaling but how A β oligomers link to presynaptic mGluR5 is not tested. Prion protein is mentioned as a possible link, and this could be tested with antibody.

6. The specificity of the CA3 versus CA1 deletion of Efr3a needs better support because the conclusions about presynaptic function are related to this. Western blots are shown in Fig. 8, but one

would expect regional protein levels are expected to include axonal as well as cell body protein. Does the blot imply Efr3a is confined to cell bodies? If so is it in or near presynaptic terminals. Further characterization with immunohistology and regional PIP2 levels may clarify the specificity of these conditional Kos.

7. The learning and memory phenotypes are differentially dependent on Efr3a. This is not explained, investigated or discussed.

8. It is remarkably that the Morris water maze were interpretable with only 4-9 mice per group. The sample size is very small for a behavioral experiment in this paradigm. In addition, most literature studies show minimal learning or memory deficit in this APP/PS1 strain at 6-7 months age. Typically, behavioral deficits occur at latter ages. The authors should address this difference from the literature.

Reviewer #3 (Remarks to the Author):

This is quite an extensive study which explores in depth the mechanisms by which A β oligomers affect presynaptic release probability at hippocampal synapses in slices and in cultured preparations. The authors show a role for decreased PIP2 levels in the axonal membrane which appears to be induced by A β -mediated activation of presynaptic mGlu5 receptors, activation of PLC to decrease release PIP2 levels and release probability. The results are compelling and of interest in the field of synaptic dysfunction in Alzheimer's disease.

There are however concerns which require additional experimental evidence and discussion.

1) As mentioned by the authors there is quite some variation in the experimental demonstration for presynaptic changes in early models of AD. There is a complex relationship between the concentration of oligomeric A β (see lines 301-319) and a presynaptic effect. The authors provide evidence for a single concentration of oligomeric A β (400nM). How was this concentration chosen, what is the range of oligomeric A β concentration for which decreased synaptic transmission is observed? This is a key issue, otherwise the experimental conditions appear very narrow and specific.

Related to this variability of effect, the authors should at least report and possibly discuss on the fact that no change in PPF was observed in AAP/PS1 mice at 6 months of age (Silva et al, Nature Communication 2016, not cited) for CA3 terminals onto CA3 cells – synapses which are normally alike CA3-CA1 synapses.

2) The authors demonstrate that bath application of oligomeric A β lead to a change in PPF (as in APP/PS1 mice), and further use long trains of stimulation to evaluate an average Pr and RRP. In none of the experiments do the authors actually show directly that synaptic transmission (amplitude of EPSCs) is decreased by any of the treatments. This should be provided whenever possible.

Related to this point (see e.g. line 118-119), the link between mEPSC frequency changes and Pr is certainly not direct. Changes in mEPSC frequency provide arguments in favour of a presynaptic mechanism, but this does not necessarily mean a change in Pr. Pr is by definition measured with relation to the effect of an action potential in a presynaptic terminal. In several instances the authors should revise sentences in which there is a confusion between changes in mEPSC frequency and Pr.

3) The behavioural experiments and associated LTP experiments are the weakest part of the article. These appear rather preliminary. What are the mechanisms by which LTP is decreased? How does this relate clearly to pre- or postsynaptic mechanisms? An impaired induction appears as the most obvious explanation, but 1) postsynaptic factors cannot be excluded; 2) if decreased induction is a key element, then increasing the length of the induction protocol may alleviate impaired LTP. The

behaviour experiments are quite isolated. The sentence "deleting Efr3a in CA3 area in APP/PS1 mice restored spatial learning but not spatial memory defects.." is for the least quite unsatisfactory. As is the link between decreased Pr and plasticity of neural ensembles in memory processing. Overall, I would be in favour of removing these unnecessary set of experiments. Including them would require a much more careful analysis.

Other points

- line 213, what is meant by "knocking-down Efr3a"? The authors should provide more explanation – as I understand it is by using of heterozygote mice. What is the relative amount of Efr3a protein expressed in these mice?
- mGlu5 is rather known to be abundant at a post-synaptic level. The authors should discuss the fact that they can exclude indirect presynaptic effects due to activation (or inhibition) of postsynaptic mGlu5.
- lines 168 to 171, 'restore' and 'rescue' are misused instead of 'prevent' or block. This paragraph could be written: "We next investigated whether blocking PLC activity prevented the presynaptic deficit... ..suggesting that blocking PLC prevented Abeta-induced impairment of Pr. Blocking Abeta-induced impairment of Pr via inhibiting PL..."
- at several occasions in the text 'that' or 'which' is omitted. The resulting sentences look a bit informal.
- line 123 – "the degree of PPF is inversely proportional to Pr". Inversely "related" would be more correct.
- lines 359-363 – The two sentences are unclear. How do they relate to the actual experiments done? See above major point #3.

Reviewer #4 (Remarks to the Author):

This is an interesting, well-written manuscript that claims to provide evidence that the synaptotoxic effect of both acutely applied and chronically produced Abeta is mediated by inhibition of presynaptic release consequent to depleting PIP2. He et al used a methodologically sound battery of tests to show that bath application of synthetic Abeta oligomers reduced mEPSC frequency and amplitude, increased PPF and reduced release probability in wild-type CA1 neurons in vitro. In contrast, although there were similar changes in CA1 of APP/PS1 mice, there was no change in mEPSC amplitude. The authors go on to report that the acute Abeta presynaptic effects were mediated by PIP2 signalling and mGluR5. The latter finding makes the manuscript potentially of interest to a broad range of readers, including clinicians, as treatment with mGluR5 antagonists is still an attractive therapeutic approach for AD. Moreover, the authors give evidence that conditional knockout in the CA3 area, but not in CA1, of Efr3a, a protein controlling PIP2 hydrolysis, restored PPF, release probability and LTP disruption by A β and in slices from APP/PS1 mice in vitro, and partially improved water-maze deficits in the transgenic mice in vivo. The key findings of the study appear to be novel. The following issues need to be addressed.

1. The authors provide no evidence that the abnormalities found in APP/PS1 mice are Abeta-dependent (e.g. are they age-dependent and/or can they be prevented by Abeta-lowering strategies?). Over-expression of APP leads to the production of many fragments. Some of these can affect presynaptic neurotransmitter release (e.g. Fanutza et al., eLife 2015; 4:e09743). Also, the role of insoluble plaques, which may appear in the brain of APP/PS1 mice at the age of 6-7 months (Garcia-Allonza et al., Neurobiology of Disease 2006; 24, 516-524), needs to be evaluated. Other factors, such as prolonged exposure to Abeta may account for the apparent difference in the effect on mEPSCs amplitude. Experiments showing that the reduction in mEPSC frequency, release probability

and increase in PPF in APP/PS1 mice are Abeta-dependent would significantly improve the manuscript.

2. The effect of MTEP on synthetic Abeta-induced mEPSC frequency and PIP2 reduction and PPF increase is interesting. However, the question arises whether mGluR5 mediates PIP2-dependent synaptic changes in APP/PS1 mice. For example, treatment with CTEP has been reported to fully reverse a deficit in APP/PS1 mice to find the hidden platform location during a probe trial in the MWM (Hamilton et al., Cell Reports 2016; 15, 1859–1865), whereas conditional knockouts of Efr3a in the CA3 area in the same transgenic animal line had no effect (Figure 7e).

3. Was the increase in PPF associated with a reduction in the epsc evoked by the first pulse? Was the input-output relationship affected by either A β or APP/PS1? For example, the mEPSC amplitude was reduced by A β , which might reduce evoked responses. Were the electrically evoked responses, RRP and PPF measured in the absence of picrotoxin, and if so does this not influence their interpretation?

4. Do the authors have direct evidence that the LTP studied here is expressed presynaptically, as implied by the Discussion?

5. Typically mGluR5 expression is found on CA1 spines, especially on the edge of synapses. If, as implied by the authors, mGluR5 are also found on CA3 axons why are these receptors apparently preferentially affected by A β ?

Reviewer #1

Many studies have established a role for soluble oligomers of amyloid-beta 42 (Ab42) in the disruption of synaptic transmission and plasticity in vitro and in vivo. A dominant hypothesis in the field is that this phenomenon may contribute to the onset of cognitive deficits in the early stages of Alzheimer's disease (AD). The bulk of those studies have focused on the negative impact of Ab42 oligomers on postsynaptic processes, including perturbation of glutamate receptor/channel trafficking and dendritic spine dynamics. This study by He et al focuses on the presynaptic impact of Ab42 oligomers through a combination of in vitro and in vivo experiments (using both pharmacology and mouse genetics), establishing a role for metabotropic glutamate receptor 5 (mGluR5) and downstream signaling through phospholipase C (PLC) and phosphatidylinositol-4,5-bisphosphate (PIP2) hydrolysis in decreasing the release probability at hippocampal synapses.

Overall, this is an interesting study that expands on previous studies from several labs indicating that PIP2 hydrolysis is a pathophysiologically relevant consequence of Ab42 oligomer increase in the brain of AD mouse models as well as in cultured hippocampal neurons. Like the other studies, the manuscript by He et al does not provide much better understanding of the molecular mechanisms occurring downstream of PIP2 hydrolysis, and specifically which PIP2 (or DAG) effectors mediate the synapse impairing actions of Ab42 oligomers. However, this study does a great job establishing a role for mGluR5 in these actions, supporting its main conclusions with elegant electrophysiologic analyses and rather compelling results. There are however several aspects that need to be addressed to make this manuscript worthy of publication in Nature Communications, in this referee's view. These are as follows:

We thank the referee for the positive opinion. We have carefully revised our manuscript and we think it is much improved.

- 1. Having established that PIP2 hydrolysis occurs downstream of Ab42 via mGluR5, it is important to determine which PLC isoforms mediate this effect presynaptically. The broad PLC inhibitor is used in this study as well as in other studies that have examined the impact of Ab42 on PIP2 metabolism, but short of elucidating the effect of PIP2 hydrolysis on molecular pathways controlling the release probability, the authors could at least determine which PLC isoform is responsible for PIP2*

hydrolysis. Granted that there are multiple PLC isoforms, it appears as though PLCbeta1 and beta4 may be good candidates. Simple shRNA experiments should allow the authors to determine whether they are the ones involved.

We thank the reviewer for his/her insightful comments and helpful suggestions. We have done RNA-interference experiments on PLC β 1 and β 4 to elucidate whether the PLC isoforms mediate oligomeric A β ₄₂-induced PIP₂ hydrolysis in hippocampal neurons as suggested by the reviewer. We found that knocking down either PLC β 1 or β 4 significantly ameliorated A β -induced suppression of PIP₂ levels in both dendrites and axons. These results indicate that both PLC β 1 and β 4 contribute to A β -induced PIP₂ hydrolysis in hippocampal neurons. We have added the data in Supplementary Figure 7, and changed the Results section (lines 207-212) accordingly.

2. This referee is not a big fan of anti-PIP2 immunostainings given the notorious artifacts reported by many investigators in the field. The in vitro studies examining PIP2 with antibodies should be complemented with studies where PH domains (fused to fluorescent proteins) are used to assess PIP2 levels microscopically. Also, they could use the same PIP2 ELISA they have employed to measure PIP2 levels in APP-PSEN1 mouse brains.

We agree with the reviewer and have measured PIP₂ levels in neuron cultures using the PIP₂ ELISA method as suggested by the reviewer. We found that oligomeric A β ₄₂ indeed suppressed PIP₂ levels in hippocampal neurons (Figure 3c). Treating hippocampal neurons that were resistant to PIP₂ hydrolysis with oligomeric A β ₄₂, however, did not reduce PIP₂ levels as revealed by the PIP₂ ELISA method (Figure 6c). These results are consistent with anti-PIP₂ immunostainings shown in Figures 3 and 6. In addition to adding panels in Figures 3 and 6, we have also revised the corresponding text in the revised manuscript.

Other comments:

3. The PIP2 depletion induced by Ab42 oligomers was already shown to be PLC-dependent in reference 40. Rather than describing their findings as novel on line 159, the authors should simply state that they confirmed previously published results.

We thank the reviewer for pointing this out. We have rephrased the sentence as “A β -induced rapid depletion of membrane PIP₂ confirms a predominant PIP₂ hydrolysis process evoked by A β oligomers⁴⁰” in lines 188-189 in the revised manuscript.

4. *The introduction focuses a lot on the physiological role of Ab42 at synapses, but the results presented in the manuscript are more related to its pathophysiological role. This referee recommends that the authors should shorten the former section and expand the latter, perhaps focusing more on the known effects of Abeta oligomers (and apoE4) on PIP2 metabolism and how levels of PIP2 phosphatase synaptojanin modulate these effects.*

We thank the reviewer for the helpful suggestions. We have shortened the overview of the physiological actions of A β (lines 60-63) and expanded the summary of PIP₂ metabolism in AD (lines 78-83) in the Introduction section of the revised manuscript.

5. *The reference “Deleterious effects of soluble amyloid- β oligomers on multiple steps of synaptic vesicle trafficking.” By Park et al. Neurobiol Dis. 2013, 55:129-39. PMID: 23523634. Should be cited and discussed, since it focuses on the presynaptic impact of Ab42 oligomers.*

We thank the reviewer for bringing this important paper to our attention. We have cited and discussed the paper in lines 135-136 and 377-379 in the revised manuscript and added the paper in the reference list.

Reviewer #2

Yang He et al explore mechanisms of synaptic dysfunction in models of AD. They report that slices from APP/PS1 transgenic at 6-7 months of age show evidence for impaired presynaptic function with decreased release probability (Pr) and decreased miniature EPP frequency. At this stage little or no change in Golgi-impregnated synaptic spines was apparent. Acute application of A β oligomer induced the same phenomenon in WT slice. The authors provide data linking this Pr decrement to activation of mGluR5 and depletion of PIP2. They utilize knockout of Efr3a as a means to prevent PIP2 depletion. They examine the functional consequence of these observations by assessing LTP in CA3-CA1 pathway and show rescue with Efr3a knockout in CA3. The deletion also rescues learning but not memory in the water maze.

These data are intriguing and contribute to the field by exploring presynaptic actions of A β oligomers to disrupt synaptic function. The hypothesis of an early presynaptic deficit mediated via mGluR5 and PIP2 would add a new dimension to mechanistic understanding of synaptic function in AD. Most work has previously described synapse loss with both pre and post elements gone, and mechanistic investigations have documented post-synaptic changes in a range of AD models. There are several aspects of the work that require further data to clarify and relate to previous work.

We thank the referee for this kind assessment of our paper. We have done additional experiments to improve the manuscript.

1. The authors emphasize that the Pr presynaptic change is seen early in APP/PS1 mice prior to synapse loss and most previous studies showing any symptoms in this line. The authors must examine earlier ages to show whether this is some developmental issue in this transgenic line or whether development is normal at 2 or 3 months age and as A β starts to accumulate the presynaptic phenotype occurs in a disease-related process.

We thank the reviewer for raising the point. In this round of revision, we mainly focused on determining whether the A β -induced presynaptic defect requires A β accumulation to certain extent. Please see also our responses to the 2nd and 3rd concerns raised by the reviewer. For this specific point, we have investigated whether the presynaptic deficit occurs earlier during development in APP/PS1 mice as suggested by the reviewer. We first recorded mEPSCs in CA1 pyramidal neurons in 4-month-old APP/PS1 mice and their wild-type littermates. We found that the frequency and amplitude of mEPSCs were not altered in APP/PS1 mice in comparison to WT mice (Supplementary Figure 1). We next elicited evoked EPSCs at the CA3-CA1 synapses and found that there were no differences in the amplitudes of EPSCs between APP/PS1 and wild-type mice at this young age (Supplementary Figure 1). Finally, we observed that the presynaptic release probability at the CA3-CA1 synapse, which was assessed by paired-pulse facilitation, was not changed in 4-month-old APP/PS1 mice compared to their wild-type littermates (Supplementary Figure 4). These data clearly show that the presynaptic defect is not a developmental issue and occurs when A β accumulates to a relatively high level (Supplementary Figure 3). We have incorporated these results in the revised manuscript.

2. *The mouse experiments utilize one AD transgenic line. Any one line may produce idiosyncratic results, so verifying the basic observation of an early presynaptic Pr issue in a second line would greatly strengthen the paper. A knock-in line would be best to study.*

We totally agree with the reviewer that verifying the AD-related presynaptic defect in other AD mouse models will significantly strengthen the manuscript. We thus tested the synaptic deficits in 5XFAD and PS1^{M146V} knock-in mice (M146V) in this round of revision. In 4-month-old 5XFAD mice with an accelerated A β production (Supplementary Fig. 2), the frequency of mEPSCs in CA1 neurons and the amplitude of SC-CA1 EPSCs were greatly reduced in comparison to APP/PS1 and WT mice of the same age (Supplementary Fig. 1a-d). By contrast, in 6-7-month-old M146V mice with a low A β production, mEPSC frequency and amplitude were not altered compared to their WT littermates (Supplementary Fig. 1e,f). Interestingly, SC-CA1 EPSCs were smaller in M146V than in control mice at 6-7-month of age, implying that an A β -independent mechanism may underlie the decreased synaptic transmission. We further found that PPF was significantly increased in 4-month-old 5XFAD (Supplementary Fig. 4a, b) mice compared to WT controls. In 6-7-month-old M146V (Supplementary Fig. 4c, d) mice, however, PPF was not changed in comparison to WT mice. We thus confirmed our observations in other AD mouse models.

3. *The authors report altered PPF in the APP/PS1 slices. This has been studied previously and others have NOT observed PPF changes (to mention just two: J Alzheimers Dis. 2018 61:195-208, Cell Death Dis. 2015 6:1676). This should be reviewed, and any potential cause discussed. The authors focus on presynaptic changes does not match well with previous work.*

We thank the reviewer for bringing these papers to our attention. In the paper by Gelman et al. (J Alzheimers Dis. 2018 61:195-208), the authors used an APP/PS1 transgenic line (APPSwe/PS1M146L) that is different from the line we used in our study (APPSwe/PS1 Δ E9). In the paper by Woo et al. (Cell Death Dis. 2015 6:1676), the authors recorded PPF in 3-month-old APPSwe/PS1 Δ E9 mice. We believe the reason why these groups have not observed PPF changes in AD mice is probably due to low soluble A β levels in their experimental conditions. We speculate that APPSwe/PS1M146L may have a low oligomeric

A β level in comparison to APPSwe/PS1 Δ E9 mice, although A β deposition begins at 3-4 months of age in these mice. We have done additional PPF experiments in 4-month-old APPSwe/PS1 Δ E9 mice and the results are similar to those obtained from 3-month-old APPSwe/PS1 Δ E9 mice by Woo et al. Soluble A β levels are low in 3-4-month-old APPSwe/PS1 Δ E9 mice (Garcia-Alloza et al., *Neurobiology of Disease* 2006 24: 516–524). We have discussed the point raised by the reviewer in the revised manuscript (lines 152-153 and 355-356) and added these papers in the reference list.

4. The biochemical measure of PIP₂ loss is in the Supplement only. It would be advantageous to show in main Figure.

We thank the reviewer for pointing this out. We have added PIP₂ ELISA data in Figures 3 and 6.

5. The pharmacology implicates mGluR5 and downstream signaling but how A β oligomers link to presynaptic mGluR5 is not tested. Prion protein is mentioned as a possible link, and this could be tested with antibody.

We have investigated whether the presynaptic target of A β , mGluR5, functions via cellular prion proteins (PrP^C). We found that in the presence of a PrP^C antibody, A β had no influence on PIP₂ levels in axons and dendrites. Activating mGluR5 directly using an agonist in the presence of the PrP^C antibody, however, still significantly reduced PIP₂ levels in neurites. These results indicate that the presynaptic PIP₂ hydrolysis induced by oligomeric A β indeed requires activation of mGluR5 in the presence of PrP^C. We have incorporated these data in Supplementary Figure 8 in the revised manuscript.

6. The specificity of the CA3 versus CA1 deletion of Efr3a needs better support because the conclusions about presynaptic function are related to this. Western blots are shown in Fig. 8, but one would expect regional protein levels are expected to include axonal as well as cell body protein. Does the blot imply Efr3a is confined to cell bodies? If so is it in or near presynaptic terminals. Further characterization with immunohistology and regional PIP₂ levels may clarify the specificity of these conditional Kos.

We thank the reviewer for raising the point. We have tried the commercial anti-Efr3a antibodies to immunostain Efr3a in both hippocampal slices and cultured hippocampal neurons. Unfortunately, we were not able to get the immunostaining to work, as the available antibodies are not suitable for immunohistochemistry/immunofluorescence studies of Efr3a. In addition to cell bodies, we believe Efr3a is also located in axons based on the following observations: 1) knocking down Efr3a significantly reduced A β -induced depletion of PIP₂ in axons (Figure 6); Efr3a interacts with GPCRs to exert its regulatory function, thus it is reasonable to reckon that Efr3a locally influences GPCRs in axons; 2) we have done Western blots multiple times in Efr3a conditional knockouts, and quantifications of Efr3a levels in CA3 or CA1 areas devoid of Efr3a in pyramidal neurons showed about 50-80% reduction in regional Efr3a levels; one of the sources of remaining Efr3a may come from axons originated from other areas as mentioned by the reviewer. We have added histograms of Efr3a levels in Figure 7 and changed the text accordingly in the revised manuscript.

7. The learning and memory phenotypes are differentially dependent on Efr3a. This is not explained, investigated or discussed.

After increasing the sample sizes in the Morris water maze experiments as suggested by the reviewer, we found that deleting Efr3a in CA3 area in APP/PS1 mice rescued both spatial learning and memory defects found in APP/PS1 mice. Please see also our response to the reviewer's 8th concern. APP/PS1 mice conditionally deleted for Efr3a in CA3 but not in CA1 areas significantly shortened the escape latency and increased the time spent in the target quadrant to the levels of WT mice in the Morris water maze test. The results can be found in Figure 9 (the original Figure 8) in the revised manuscript.

8. It is remarkably that the Morris water maze were interpretable with only 4-9 mice per group. The sample size is very small for a behavioral experiment in this paradigm. In addition, most literature studies show minimal learning or memory deficit in this APP/PS1 strain at 6-7 months age. Typically, behavioral deficits occur at latter ages. The authors should address this difference from the literature.

We thank the reviewer for pointing this out. We accept the criticism that the sample size of 4 is not sufficient to draw a clear conclusion in the Morris water maze test. Indeed, after

increasing the sample sizes to 7-9, we found that APP/PS1 mice conditionally deleted for Efr3a in CA3 but not in CA1 areas significantly increased the time spent in the target quadrant in the memory retrieval phase of the Morris water maze test. This was not observed when the sample sizes were small in the first submission of the manuscript. Based on our results and other reports (e.g. Trinchese et al., J Clin Invest. 2008; Wang et al., Behav. Brain Res. 2012), the spatial learning defect in APP/PS1 mice is generally began to show since the age of 6 months, although the earliest deficit can be found in 4-5-month-old APP/PS1 mice (Kummer et al., J Neurosci. 2014). We were able to reliably separate 6-7-month-old APP/PS1 mice from their wild-type littermates in the escape latency and target time plots (Figure 9), although the separations would be more apparent if we had performed the behavioral test in older mice as mentioned by the reviewer. Another reason we performed the behavioral test in 6-7-month-old mice was to match the electrophysiological data. Although we are capable of recording synaptic responses in 8-month-old mice, electrophysiological recordings performed in 6-7-month-old mice were much more stable. We have changed the text in lines 322-325 in the revised manuscript.

Reviewer #3

This is quite an extensive study which explores in depth the mechanisms by which A β oligomers affect presynaptic release probability at hippocampal synapses in slices and in cultured preparations. The authors show a role for decreased PIP2 levels in the axonal membrane which appears to be induced by A β -mediated activation of presynaptic mGlu5 receptors, activation of PLC to decrease release PIP2 levels and release probability. The results are compelling and of interest in the field of synaptic dysfunction in Alzheimer's disease.

We thank the reviewer for his/her kind words and helpful comments. We have carefully revised our manuscript and we think it is much improved.

There are however concerns which require additional experimental evidence and discussion.

1) As mentioned by the authors there is quite some variation in the experimental demonstration for presynaptic changes in early models of AD. There is a complex relationship between the concentration of oligomeric A β (see lines 301-319) and a presynaptic effect. The authors provide evidence for a single concentration of oligomeric A β (400nM). How was this concentration chosen,

what is the range of oligomeric A β concentration for which decreased synaptic transmission is observed? This is a key issue, otherwise the experimental conditions appear very narrow and specific.

Related to this variability of effect, the authors should at least report and possibly discuss on the fact that no change in PPF was observed in AAP/PS1 mice at 6 months of age (Silva et al, Nature Communication 2016, not cited) for CA3 terminals onto CA3 cells – synapses which are normally alike CA3-CA1 synapses.

We thank the reviewer for the insightful comments. In the first submission of the manuscript, we actually tried how low concentration oligomeric A β influenced mEPSCs recorded in CA1 pyramidal neurons. We found that 100 nM A β did not affect mEPSC amplitude and frequency (data not shown). We gradually increased A β concentration and found that at a concentration of 400 nM oligomeric A β was able to significantly reduce mEPSC frequency. We thus chose 400 nM oligomeric A β to explore its presynaptic effect. In this round of revision, as suggested by the referee, we have done additional experiments to explore the role of oligomeric A β at various concentrations in modulating hippocampal synaptic transmission. The results are summarized in Supplementary Figure 3 in the revised manuscript. We found that 20 nM oligomeric A β significantly increased the amplitude of EPSCs recorded at the CA3-CA1 synapses. We observed a slight decrease in the amplitude of EPSCs when 100 nM oligomeric A β was applied in the bath solution. Oligomeric A β at a concentration of 400 nM, however, significantly reduced EPSC amplitude. A β fibrils (400 nM), on the other hand, had a minimal effect on EPSC amplitude. We thus chose 400 nM A β oligomers to explore the mechanism underlying A β -induced synaptic deficits.

We thank the referee for bringing this important paper to our attention. We notice that the PPF ratio is nearly 2 at the CA3-CA3 synapse in both 6-month-old wild-type and APP/PS1 mice, suggesting that the release probability is extremely low at this synapse. Pathological level of A β thus may not be able to further decrease the release probability. We have cited and discussed the paper in lines 379-381 in the revised manuscript.

2) The authors demonstrate that bath application of oligomeric A β lead to a change in PPF (as in APP/PS1 mice), and further use long trains of stimulation to evaluate an average Pr and RRP. In

none of the experiments do the authors actually show directly that synaptic transmission (amplitude of EPSCs) is decreased by any of the treatments. This should be provided whenever possible.

Related to this point (see e.g. line 118-119), the link between mEPSC frequency changes and Pr is certainly not direct. Changes in mEPSC frequency provide arguments in favour of a presynaptic mechanism, but this does not necessarily mean a change in Pr. Pr is by definition measured with relation to the effect of an action potential in a presynaptic terminal. In several instances the authors should revise sentences in which there is a confusion between changes in mEPSC frequency and Pr.

We thank the reviewer for the helpful comment. As suggested by the reviewer, we have done additional experiments to show that bath application of oligomeric A β significantly suppressed evoked EPSCs at the synapses between the Schaffer collateral and CA1 pyramidal neurons (Figure 1). In addition, we have recorded evoked EPSCs in 6-7-month-old APP/PS1 mice and wild-type controls (Figure 1), 4-month-old APP/PS1 and 5XFAD mice and wild-type controls (Supplementary Figure 1), and 6-7-month-old M146V and control mice (Supplementary Figure 1). In 6-7-month-old APP/PS1 and 4-month-old 5XFAD mice with a high level of A β production, we showed that the evoked EPSCs were strongly decreased. These data are consistent with A β -induced reduction of mEPSC frequency and presynaptic release probability (*Pr*). Interestingly, we found that the evoked EPSCs were also decreased in 6-7-month-old M146V mice, although these mice exhibited a minimal increase in A β production. Although we do not know exactly what causes this reduction in M146V mice, we are able to rule out *Pr* reduction in these mice (Supplementary Figure 4).

We agree with the reviewer that changes in mEPSC frequency are not always associated with *Pr* alterations. We have carefully revised the text to avoid such confusion.

3) The behavioural experiments and associated LTP experiments are the weakest part of the article. These appear rather preliminary. What are the mechanisms by which LTP is decreased? How does this relate clearly to pre- or postsynaptic mechanisms? An impaired induction appears as the most obvious explanation, but 1) postsynaptic factors cannot be excluded; 2) if decreased induction is a key element, then increasing the length of the induction protocol may alleviate impaired LTP. The behaviour experiments are quite isolated. The sentence "deleting Efr3a in CA3 area in APP/PS1

mice restored spatial learning but not spatial memory defects.." is for the least quite unsatisfactory. As is the link between decreased Pr and plasticity of neural ensembles in memory processing. Overall, I would be in favour of removing these unnecessary set of experiments. Including them would require a much more careful analysis.

We thank the reviewer for these insightful comments. We accept the criticism that it is premature to link decreased Pr to impaired LTP. In this round of revision, we have made a lot of efforts to establish the link (Figures 8 and 9). Firstly, we used a protocol (3 bursts of 20 pulses at 100 Hz separated by 1.5 s) to induce LTP that had a robust presynaptic element as previously described (Emptage et al., Neuron, 2003). Secondly, we compared PPF ratio before and after this train stimulation and found that decreased PPF ratio was associated with LTP induction. Thirdly, we used a pharmacological method to block mGluR5 and found that mGluR5 blockade prevented oligomeric A β -induced LTP impairment. Lastly, we genetically reduced the responsiveness of presynaptic GPCRs to A β and found that A β no longer decreased LTP as a result. Although we are not able to exclude postsynaptic factors that are involved in A β -induced LTP impairment, especially in the case of a common LTP induction protocol we used in the first submission of the manuscript, these results clearly show that A β -induced suppression of Pr represents a major mechanism by which LTP is decreased.

We agree it is intriguing to see that the spatial learning and memory phenotypes are differentially dependent on presynaptic Efr3a. It turns out that this is due to the small sample sizes we used in the behavioral tests. After increasing the sample sizes in this round of revision, we found that deleting Efr3a in CA3 area in APP/PS1 mice rescued both spatial learning and memory defects found in APP/PS1 mice (Figure 9). Please see also our responses to reviewer 3's 7th and 8th points.

Other points

- line 213, what is meant by "knocking-down Efr3a"? The authors should provide more explanation – as I understand it is by using of heterozygote mice. What is the relative amount of Efr3a protein expressed in these mice?

Yes, it is meant by using of Efr3a heterozygote mice. The relative amount of Efr3a protein expressed in these mice is about 40% of the wild-type mice. We have changed the text in line

255 to specifically mention the heterozygote mice and added western blots of Efr3a in Supplementary Figure 9 in the revised manuscript.

- mGlu5 is rather known to be abundant at a post-synaptic level. The authors should discuss the fact that they can exclude indirect presynaptic effects due to activation (or inhibition) of postsynaptic mGlu5.

As suggested by the reviewer, we have discussed in the revised manuscript that we can exclude indirect presynaptic effects due to A β -induced activation of postsynaptic mGluR5. When the responsiveness of mGluR5 was reduced specifically in CA3 areas, A β oligomers were not able to alter the presynaptic release probability at the CA3-CA1 synapse, indicating that postsynaptic mGluR5 was not involved in this A β -induced presynaptic deficit.

- lines 168 to 171, 'restore' and 'rescue' are misused instead of 'prevent' or block. This paragraph could be written: "We next investigated whether blocking PLC activity prevented the presynaptic deficit... ..suggesting that blocking PLC prevented Abeta-induced impairment of Pr. Blocking Abeta-induced impairment of Pr via inhibiting PL..."

We thank the reviewer for the helpful suggestion. We have changed the sentences as suggested by the reviewer.

- at several occasions in the text 'that' or 'which' is omitted. The resulting sentences look a bit informal.

As suggested by the reviewer, we have added "that" or "which" in lines 129, 186, 201, 229, 231, 244, 250, 277, 303, and 321 in the revised manuscript.

- line 123 – "the degree of PPF is inversely proportional to Pr". Inversely "related" would be more correct.

As suggested by the reviewer, we have replaced "proportional" to "related" in line 148 (original line 123).

- lines 359-363 – The two sentences are unclear. How do they relate to the actual experiments done? See above major point #3.

Based on our new data of LTP (Figures 8 and 9), we have expanded the discussion on page 20 (lines 420-422) in the revised manuscript.

Reviewer #4

This is an interesting, well-written manuscript that claims to provide evidence that the synaptotoxic effect of both acutely applied and chronically produced Abeta is mediated by inhibition of presynaptic release consequent to depleting PIP2. He et al used a methodologically sound battery of tests to show that bath application of synthetic Abeta oligomers reduced mEPSC frequency and amplitude, increased PPF and reduced release probability in wild-type CA1 neurons in vitro. In contrast, although there were similar changes in CA1 of APP/PS1 mice, there was no change in mEPSC amplitude. The authors go on to report that the acute Abeta presynaptic effects were mediated by PIP2 signalling and mGluR5. The latter finding makes the manuscript potentially of interest to a broad range of readers, including clinicians, as treatment with mGluR5 antagonists is still an attractive therapeutic approach for AD. Moreover, the authors give evidence that conditional knockout in the CA3 area, but not in CA1, of Efr3a, a protein controlling PIP2 hydrolysis, restored PPF, release probability and LTP disruption by Aβ and in slices from APP/PS1 mice in vitro, and partially improved water-maze deficits in the transgenic mice in vivo. The key findings of the study appear to be novel.

We thank the referee for the positive opinion and kind assessment of our paper.

The following issues need to be addressed.

1. *The authors provide no evidence that the abnormalities found in APP/PS1 mice are Abeta-dependent (e.g. are they age-dependent and/or can they be prevented by Abeta-lowering strategies?). Over-expression of APP leads to the production of many fragments. Some of these can affect presynaptic neurotransmitter release (e.g. Fanutza et al., eLife 2015; 4:e09743). Also, the role of insoluble plaques, which may appear in the brain of APP/PS1 mice at the age of 6-7 months (Garcia-Allonza et al., Neurobiology of Disease 2006; 24, 516-524), needs to be evaluated. Other factors, such as prolonged exposure to Abeta may account for the apparent difference in the effect on mEPSCs amplitude. Experiments showing that the reduction in mEPSC frequency, release probability and increase in PPF in APP/PS1 mice are Abeta-dependent would significantly improve the manuscript.*

We thank the reviewer for his/her insightful comments and helpful suggestions. To address if the presynaptic abnormality found in APP/PS1 mice are indeed Aβ-dependent, we

first tested if the decreased presynaptic release probability was developmentally regulated as suggested by the reviewer. We found that the frequency and amplitude of mEPSCs in CA1 pyramidal neurons (Supplementary Figure 1) and PPF at the CA3-CA1 synapse (Supplementary Figure 4) were not altered in 4-month-old APP/PS1 mice in comparison to wild-type controls. We next investigated the synaptic defects in response to synthetic A β oligomers at different concentrations and confirmed that the presynaptic action of A β oligomers requires a significantly elevated oligomeric A β level (Supplementary Figure 3). Finally, we tested the synaptic deficits in 5XFAD and M146V mice (Supplementary Figures 1 and 2). In 4-month-old 5XFAD mice, a mouse model of AD that shows accelerated A β production, we observed that mEPSC frequency was reduced and PPF was increased. In 6-month-old M146V mice, a presenilin 1 mutant mouse that exhibits minimal increase in A β at this age, mEPSC frequency and PPF ratio were not altered in comparison to their wild-type littermates. These new results demonstrate that the reduction in mEPSC frequency and increase in PPF in AD mouse models are clearly A β -dependent.

2. *The effect of MTEP on synthetic Abeta-induced mEPSC frequency and PIP2 reduction and PPF increase is interesting. However, the question arises whether mGluR5 mediates PIP2-dependent synaptic changes in APP/PS1 mice. For example, treatment with CTEP has been reported to fully reverse a deficit in APP/PS1 mice to find the hidden platform location during a probe trial in the MWM (Hamilton et al., Cell Reports 2016; 15, 1859–1865), whereas conditional knockouts of Efr3a in the CA3 area in the same transgenic animal line had no effect (Figure 7e).*

It turns out that no improvement in the probe trial test for APP/PS1 mice conditionally deleted for Efr3a in the CA3 area is due to the small sample sizes we used in the original behavioral experiments. Please see also our response to reviewer 2's 7th point. After increasing the sample sizes, we found that APP/PS1 mice conditionally deleted for Efr3a in CA3 but not in CA1 areas reversed their deficits of finding the hidden platform in the memory retrieval phase of the Morris water maze test. The results can be found in Figure 9 (the original Figure 8) in the revised manuscript.

3. *Was the increase in PPF associated with a reduction in the EPSC evoked by the first pulse? Was the input-output relationship affected by either A β or APP/PS1? For example, the mEPSC amplitude was reduced by A β , which might reduce evoked responses. Were the electrically evoked responses, RRP and PPF measured in the absence of picrotoxin, and if so does this not influence their interpretation?*

We thank the reviewer for the insightful comments. We have done additional experiments to explore if the PPF ratio is affected by the magnitude of the first EPSC. We found that the PPF ratio did not change significantly when we altered the stimulation intensity, indicating that A β -induced suppression of EPSC amplitude had minimal influence on the presynaptic release probability. We have incorporated these data in the revised Figure 2. We have also recorded the evoked responses at the CA3-CA1 synapses as suggested by the reviewer. We found that pathological level of A β caused a significant decrease in the evoked responses at the CA3-CA1 synapses. In 6-7-month-old APP/PS1 mice, the average amplitudes of evoked EPSCs at the CA3-CA1 synapses were also significantly reduced in comparison to wild-type mice of the same age (Figure 1). Please see also our response to reviewer 3's 2nd concern. We are sorry that we did not provide detailed information of the recording conditions when we elicited evoked EPSCs at the CA3-CA1 synapses in the first version of the manuscript. We actually recorded evoked EPSCs in the presence of 10 μ M bicuculline to inhibit inhibitory synaptic responses. We have added this information in the Method section of the revised manuscript.

4. *Do the authors have direct evidence that the LTP studied here is expressed presynaptically, as implied by the Discussion?*

In the first submission of the manuscript, we used a standard protocol to induced LTP at the CA3-CA1 synapses. Two consecutive tetanic stimuli (frequency = 100 Hz, duration = 1 s) separated by 20 s were delivered to induce LTP. We believe LTP induced by this protocol contains a presynaptic component, as decreasing A β -induced depletion of PIP₂ in CA3 area restored LTP at the CA3-CA1 synapses in APP/PS1 mice (the original Figure 8 in the first submission of the manuscript). To fully establish that the A β -induced presynaptic defect influences synaptic plasticity, we used a protocol (3 bursts of 20 pulses at 100 Hz separated

by 1.5 s) to induce LTP that contains a robust presynaptic component as previously described (Emptage et al., Neuron, 2003). In Figure 8 of the revised manuscript, we show that this LTP induction protocol causes a significant decrease in the PPF ratio, indicating that the presynaptic release probability is enhanced following LTP induction. In the presence of oligomeric A β , however, the PPF ratio does not change significantly following LTP induction. We thus provide direct evidence that oligomeric A β suppresses presynaptically expressed LTP in the hippocampus. Please see also our response to reviewer 3's 3rd concern.

5. *Typically mGluR5 expression is found on CA1 spines, especially on the edge of synapses. If, as implied by the authors, mGluR5 are also found on CA3 axons why are these receptors apparently preferentially affected by A β ?*

We believe both presynaptic and postsynaptic metabotropic glutamate receptors 5 (mGluR5) are targeted by oligomeric A β . Using the PIP₂ level as a readout for mGluR5 activation, however, we showed that oligomeric A β (Figure 3) and DHPG (Figure 5) depleted PIP₂ to a greater level in axons than in dendrites. Although we do not know exactly what causes this differential regulation of PIP₂ in neurites, we nevertheless speculate that this may be due primary to a combination of enhanced mGluR5 receptor responsiveness and downstream process efficacy in axons, because mGluR5 density is higher in dendrites than in axons (Figure 5a,b). Of course, solving this issue requires a significant amount of efforts and a separate research project. On the other hand, we do believe activation of postsynaptic mGluR5 by oligomeric A β has a minimal effect on synaptic transmission in the hippocampus, as blocking mGluR5 did not restore A β -induced inhibition of mEPSC amplitude. We have quantified mGluR5 fluorescence intensities in axons and neurons in Figure 5 and discussed this issue in lines 398-403 in the revised manuscript.

Reviewers' comments:

Reviewer #2 (Remarks to the Author):

Appropriately revised.

Reviewer #3 (Remarks to the Author):

The authors have very significantly improved their manuscript. A few additional comments.

1) The authors have focussed on the SC-CA1 synapse. I still strongly think that the authors should more clearly mention that this may not be true at all synapse types. In particular, the fact that CA3-CA3 synapses are not altered at a presynaptic level in 6 months old APP/PS1 mice should be more clearly mentioned, for instance line 153. It is not correct to say that CA3-CA3 have an extremely low Pr (see for instance Holderith et al, 2012, or Guzman et al, 2016). It is quite variable but can rise up to .53. Hence the authors do not have sufficient arguments to claim that an "extremely low Pr" can explain synaptic specificity of the effect they see.

2) the sentence lines 61-63 is difficult to understand and should be rephrased.

3) line 259. "knocking-down Efr3a in APP/PS1" is still unclear.

Reviewer #4 (Remarks to the Author):

Although the revised version contains a significant amount of new experimental data which certainly improves the manuscript, the link between synthetic Abeta effects and the AD mouse models appears even weaker now. In my opinion the following major points need to be considered:

1

The authors now report evidence that presynaptic changes in APP/PS1 mice are age-dependent and compare them with two other transgenic mouse AD models. They conclude that the changes in the transgenic mouse AD models are A β -dependent but the data are not strong.

The Western blot in Supplementary Figure 2 does not make sense. 6E10 is an antibody to human A β (e.g. see Biologend source material) yet bands labeled 'A β ' are presented on the blots of the control B6 (lane 1) and S129 (lane 7) WT mice lacking human A β . Furthermore, the control protein β -actin band varies greatly between mice. This is not acceptable and doesn't provide very convincing evidence of an age-dependent or genotype-mediated A β accumulation in the mice studied.

Strangely, the authors state that "Excitatory synaptic transmission is normal in 6-month-old M146V mice" in the legend for Supplementary Figure 1, whereas they show that the i/o curve was significantly reduced. There is no satisfactory evidence regarding the A β -dependence of the reduction in baseline synaptic transmission in the mice, or indeed the 5XFAD (supplementary Figure 1) or APP/PS1 (Figure 1) mice.

To reinforce their message the authors now report that acute application of A β has a concentration-dependent effect on baseline transmission and concluded that the reduction of the EPSC was not prevented by MTEP or U73122 (whereas the reduction in mEPSC frequency was). This implies that the

A β -triggered baseline decrease is mediated by a different mechanism from the A β -triggered reduction in probability of release, which is somewhat surprising. For example, DHPG is known to cause an mGluR-dependent baseline decrease that has been associated with a reduction in presynaptic release.

Were the measurements of baseline, I/O curves, PPF and release probability taken at the same time relative to the start of A β application? The decline in baseline caused by A β (400 nM) appears gradual and didn't appear to have plateaued by 20 min (~50%, see Supplementary Figure 3). In the presence of MTEP or U73122 the A β -induced decline seems less (~75% at 20 min, see Figure 4 and 5). Why wasn't the A β alone group compared directly with these groups? Was the afferent nerve volley reduced at this concentration of A β ? Did knocking out Efr3a ameliorate the acute A β -induced baseline depression?

The ~50% increase in baseline caused by 20 nM A β is very dramatic (Supplementary Figure 3). What is the basis of this increase? Have any previous studies reported this phenomenon and can they throw light on the mechanism?

2

The authors have increased the ns for the behavioral study in Efr3a mice but don't appear to address the issue of the mGluR5-dependence of the synaptic changes in APP/PS mice.

3

Would not the large reduction in baseline transmission in many experimental groups be expected to negatively impact the magnitude of the effective stimulus during induction of LTP? Was any attempt made to compensate for the reduction in baseline in these groups?

The method given for LTP induction needs changing to the new protocol in the main text (p23).

4

Why don't the presumably "representative" fEPSP pre-HFS traces of PPF in Figure 9 not parallel the described changes in PPF of the EPSC? If anything, the two cases where A β was reported to increase PPF of the EPSC (A β alone and in CA1Efr mice), now appear to reduce PPF of the fEPSP compared with relevant groups (e.g. WT control and CA3Efr mice, respectively).

Looking more closely at some of the representative traces of PPF of the EPSC, the control magnitude seems very low (see e.g. Figure 7b and g).

Reviewer #3

The authors have very significantly improved their manuscript. A few additional comments.

1) The authors have focused on the SC-CA1 synapse. I still strongly think that the authors should more clearly mention that this may not be true at all synapse types. In particular, the fact that CA3-CA3 synapses are not altered at a presynaptic level in 6 months old APP/PS1 mice should be more clearly mentioned, for instance line 153. It is not correct to say that CA3-CA3 have an extremely low Pr (see for instance Holderith et al, 2012, or Guzman et al, 2016). It is quite variable but can rise up to .53. Hence the authors do not have sufficient arguments to claim that an "extremely low Pr" can explain synaptic specificity of the effect they see.

We thank the reviewer for his/her kind words and helpful comments. We agree without sufficient data we can not claim CA3-CA3 synapses have a low release probability. We have clearly mentioned this in lines 420-424 in the revised manuscript as suggested by the reviewer.

2) the sentence lines 61-63 is difficult to understand and should be rephrased.

We have rephrased the sentence to “Physiological level of A β may augment Pr via increasing presynaptic Ca²⁺ by promoting presynaptic amyloid precursor protein (APP) homodimerization, activating exocytotic Ca²⁺ channels, and regulating presynaptic α 7 nicotinic acetylcholine receptors” in lines 60-63 in the revised manuscript.

3) line 259. "knocking-down Efr3a in APP/PS1" is still unclear.

We have changed “knocking-down Efr3a in APP/PS1 mice” to “halving Efr3a copy number in APP/PS1 mice” (in line 296 in the revised manuscript) to make the sentence clearer.

Reviewer #4

Although the revised version contains a significant amount of new experimental data which certainly improves the manuscript, the link between synthetic A β effects and the AD mouse models appears even weaker now. In my opinion the following major points need to be considered:

1

The authors now report evidence that presynaptic changes in APP/PS1 mice are age-dependent and compare them with two other transgenic mouse AD models. They conclude that the changes in the transgenic mouse AD models are A β -dependent but the data are not strong.

The Western blot in Supplementary Figure 2 does not make sense. 6E10 is an antibody to human A β (e.g. see Biolegend source material) yet bands labeled 'A β ' are presented on the blots of the control B6 (lane 1) and S129 (lane 7) WT mice lacking human A β . Furthermore, the control protein β -actin band varies greatly between mice. This is not acceptable and doesn't provide very convincing evidence of an age-dependent or genotype-mediated A β accumulation in the mice studied.

We thank the reviewer for pointing this out. Indeed, 6E10 only recognizes human A β , although some labs reported non-specific binding using this antibody (see Teich et al., 2013). We put a lot of efforts to uncover why there were so many cross-reactions on the blots in this round of revision. We found that the secondary antibody we used (goat anti-mouse IgG, Thermo Scientific, 31430) generated non-specific bands mainly at 60 KDa and 25 KDa even without a prior incubation with 6E10 (see Figure A below). In addition, blocking with BSA caused a great amount of non-specific binding (see Figure B below). When blocking with 5% non-fat milk in addition to using a different secondary antibody (goat anti-mouse IgG, Multi Sciences, 70-GAM007), non-specific binding was greatly reduced (see Figure C below). We thus used this optimized method to detect APP and A β oligomers and the results can be found in Supplementary Figure 2 in the revised manuscript.

Figure A Incubation brain samples with a secondary antibody (Thermo Scientific, 31430) alone generates non-specific binding.

Figure B Blocking with 5% BSA generates a great amount of non-specific binding.

Figure C Blocking with 5% non-fat milk and incubating with a different secondary antibody (Multi Sciences, 70-GAM007) reduced non-specific binding.

Strangely, the authors state that “Excitatory synaptic transmission is normal in 6-month-old M146V mice” in the legend for Supplementary Figure 1, whereas they show that the i/o curve was significantly reduced. There is no satisfactory evidence regarding the A β -dependence of the reduction in baseline synaptic transmission in the mice, or indeed the 5XFAD (supplementary Figure 1) or APP/PS1 (Figure 1) mice.

We thank the referee for this comment. This is indeed a mistake to claim that excitatory synaptic transmission is normal in 6-month-old M146V mice. What we meant by excitatory synaptic transmission was actually presynaptic release probability. The central topic of this study is to link decreased presynaptic release probability with elevated A β oligomers. Although the evoked EPSCs are smaller in 6-month-old M146V mice than in WT mice of the same age, the release probability of the SC-CA1 synapse is not reduced in M146V mice as implicated by unaltered PPF and mEPSC frequency in comparison to WT mice. This observation is consistent with a low A β production in M146V mice. We agree with the referee that the overall reduction in baseline transmission may not be A β -dependent in M146V mice, as M146V mutation in presenilin 1 is known to dephosphorylate postsynaptic GluA1 subunit in a calcineurin-dependent manner (see Kim, Violette, Ziff, 2015), potentially leading to

decreased synaptic transmission in these mice. We have changed the legend of Supplementary Figure 1 in the revised manuscript.

We do believe the A β -induced reduction in transmitter release contributes to the decreased baseline synaptic transmission in APP/PS1 mice. Conditionally knocking out Efr3a in CA3 area to increase release probability greatly enhanced the reduced synaptic transmission at the CA3-CA1 synapse in APP/PS1 mice (Supplementary Figure 15). Furthermore, we have done additional experiments to explore if blocking A β influences PPF and baseline synaptic transmission in APP/PS1 mice as suggested by the editors (Figure 3 and Supplementary Figure 5). We first used a well-established method to decrease A β production by blocking γ -secretase with LY-411575 in APP/PS1 mice (see Cirrito et al., 2003; Lanzet et al., 2004; Abramowski et al., 2008; Harris et al., 2010). Cirrito et al. (2003) nicely showed that LY-411575 reduced interstitial fluid A β by 80-90% in APP transgenic mice using an in vivo microdialysis method. Treatment with LY-411575 significantly increased mEPSC frequency, partially but considerably restored evoked EPSCs, and substantially decreased PPF in 6-7-month-old APP/PS1 mice (Supplementary Figure 5). To directly block A β , we used an anti-A β approach in APP/PS1 slices as suggested by the editors. We incubated hippocampal slices in ACSF containing 6E10 or control ACSF and found that only long-term 6E10 treatment (> 5 h) strongly increased mEPSC frequency and evoked EPSC amplitude and suppressed PPF in APP/PS1 mice (Figure 3), indicating that blocking A β with 6E10 indeed restored the presynaptic defect in these mice. Although this long-term maintenance of hippocampal slices in vitro caused a rundown in the baseline transmission in both WT and APP/PS1 mice, the effect of blocking A β with 6E10 on restoring synaptic transmission in APP/PS1 mice to the level in WT mice was remarkable. These data confirm presynaptic release probability is A β -dependent. We have incorporated these results in lines 166-189 in the revised manuscript.

To reinforce their message the authors now report that acute application of A β has a concentration-dependent effect on baseline transmission and concluded that the reduction of the EPSC was not prevented by MTEP or U73122 (whereas the reduction in mEPSC frequency was).

This implies that the A β -triggered baseline decrease is mediated by a different mechanism from the A β -triggered reduction in probability of release, which is somewhat surprising. For example, DHPG is known to cause an mGluR-dependent baseline decrease that has been associated with a reduction in presynaptic release.

We thank the reviewer for raising the point. What we described in the previous revision regarding the effect of U73122 on A β -triggered EPSC reduction is: “In the presence of U73122, oligomeric A β ₄₂ still **slightly suppressed the amplitude of SC-CA1 EPSCs”. A β -induced suppression of EPSCs was actually partially prevented by U73122 or MTEP, indicating that the A β -induced reduction in probability of release contributes to the A β -triggered baseline decrease to a certain extent. These results are in line with previous studies that showed that a reduction in presynaptic release causes a decrease in evoked responses. We have respectively modified the main text in lines 228-229 and 267 to clarify the effects of U73122 and MTEP in the revised manuscript. As suggested by the reviewer, we have performed additional experiments and incorporated the time course of A β -induced suppression of EPSCs in Figures 5 and 6 in the revised manuscript. Comparing the A β alone group to the A β + U73122 or A β + MTEP group clearly shows A β -induced suppression of EPSCs was partially prevented by U73122 or MTEP.**

Were the measurements of baseline, I/O curves, PPF and release probability taken at the same time relative to the start of A β application? The decline in baseline caused by A β (400 nM) appears gradual and didn't appear to have plateaued by 20 min (~50%, see Supplementary Figure 3). In the presence of MTEP or U73122 the A β -induced decline seems less (~75% at 20 min, see Figure 4 and 5). Why wasn't the A β alone group compared directly with these groups? Was the afferent nerve volley reduced at this concentration of A β ? Did knocking out Efr3a ameliorate the acute A β -induced baseline depression?

A β -induced reduction in evoked EPSCs normally reaches to a plateau 15 - 20 min after A β treatment (see Figures 5, 6, and 7 and Supplementary Figure 3). Thus, we started to record A β -induced changes in mEPSCs, evoked responses, and short- and long-term synaptic plasticity at about 20 min after A β incubation. A β had no effect on the presynaptic volley

when we stimulated the Schaffer collateral. We have incorporated these in Supplementary Methods in the current revision. As suggested by the reviewer, we have done additional experiments and added A β alone data in Figures 5 and 6 (i.e. the original Figures 4 and 5). We have also incorporated new data in Figure 7 to show that halving *Efr3a* copy number ameliorated A β -induced baseline suppression.

The ~50% increase in baseline caused by 20 nM A β is very dramatic (Supplementary Figure 3). What is the basis of this increase? Have any previous studies reported this phenomenon and can they throw light on the mechanism?

Increasing A β level moderately has been previously shown to enhance baseline transmission substantially. For example, increasing extracellular A β by inhibiting A β degradation increases basal synaptic transmission and decreases short-term synaptic facilitation in CA3-CA1 synapses (Abramov et al., 2009). Applying 1-5 nM A β oligomers intracellularly significantly augments baseline EPSCs in CA3-CA1 synapses (Whitcomb et al., 2015). A β -induced elevation of baseline EPSCs may engage accompanying presynaptic and postsynaptic changes. Enhanced vesicle release and rapid postsynaptic insertion of GluA1 have been linked to A β -induced baseline augmentation. We have discussed this A β -induced increase in synaptic transmission in lines 402-405 in the revised manuscript.

2

*The authors have increased the ns for the behavioral study in *Efr3a* mice but don't appear to address the issue of the mGluR5-dependence of the synaptic changes in APP/PS mice.*

We thank the reviewer for the helpful comment. We have done additional experiments to investigate if blocking mGluR5 changes the synaptic transmission in APP/PS1 mice as suggested by the reviewer. We incubated hippocampal slices prepared from 6-7-month-old APP/PS1 mice with MTEP-containing ACSF for at least 3 hours and found that this MTEP treatment significantly increased mEPSC frequency and evoked EPSC amplitude and decreased PPF in comparison to maintaining control slices in vehicle-containing ACSF for the same amount of time. Short-term MTEP treatment did not change the synaptic

transmission in APP/PS1 mice (data not shown). Although this long-term maintenance of hippocampal slices prepared from animals at 6-7 months of age *in vitro* was technically challenging and decreased the baseline transmission to a certain extent, MTEP treatment significantly restored synaptic deficits in APP/PS1 mice. We have incorporated the results in Supplementary Figure 11 in the revised manuscript.

3

Would not the large reduction in baseline transmission in many experimental groups be expected to negatively impact the magnitude of the effective stimulus during induction of LTP? Was any attempt made to compensate for the reduction in baseline in these groups?

We thank the reviewer for bringing this out. We actually adjusted the amplitude of the baseline fEPSPs in control groups to match that in groups with reduced fEPSPs. We have incorporated this information in the Supplementary methods in the revised manuscript.

The method given for LTP induction needs changing to the new protocol in the main text (p23).

We thank the reviewer for pointing this out. We have changed the LTP induction protocol in lines 534-535 in the revised manuscript.

4

Why don't the presumably "representative" fEPSP pre-HFS traces of PPF in Figure 9 not parallel the described changes in PPF of the EPSC? If anything, the two cases where A β was reported to increase PPF of the EPSC (A β alone and in CA1Efr mice), now appear to reduce PPF of the fEPSP compared with relevant groups (e.g. WT control and CA3EFfr mice, respectively).

The superimposed fEPSP traces in the original Figure 9 (Figure 10 in the current revision) are not traces of PPF. We just superimposed baseline fEPSP traces with fEPSPs after LTP induction to show the degree of LTP.

Looking more closely at some of the representative traces of PPF of the EPSC, the control magnitude seems very low (see e.g. Figure 7b and g).

These EPSC traces in the original Figure 7b and g (Figure 8b and g in the current revision) are actually comparable in size to the traces in Figure 7k and o. Although the scale bars in Figure 7b and g are 50 pA, the lengths of these bars are almost half of lengths of bars (100 pA) in Figure 7k and o. Furthermore, the magnitude of the first EPSC does not affect the degree of PPF drastically as illustrated in Figure 2.

REVIEWERS' COMMENTS:

Reviewer #3 (Remarks to the Author):

The authors have made corrections in the text to answer my remaining concerns. The manuscript contains a great deal of interesting information to be published.

Reviewer #4 (Remarks to the Author):

The authors have made a valiant effort to meet my concerns. I have no further comments. Thanks.

Reviewer #3

The authors have made corrections in the text to answer my remaining concerns. The manuscript contains a great deal of interesting information to be published.

We thank the reviewer for the enthusiastic comment. We are grateful to the reviewer's constructive comments during the reviewing process.

Reviewer #4

The authors have made a valiant effort to meet my concerns. I have no further comments. Thanks.

We thank the reviewer for the kind feedback of the previous revision of the manuscript. We are grateful to the reviewer's constructive comments during the reviewing process.